# *Vibrio cholerae's* ToxRS bile sensing system

**Nina Gubensäk[1]\*, Theo Sagmeister[1], Christoph Buhlheller[1], Bruno Di Geronimo[2], Gabriel E Wagner[3,4], Lukas Petrowitsch[1], Melissa A Gräwert[5], Markus Rotzinger[3], Tamara M Ismael Berger[1], Jan Schäfer[6], Isabel Usón[7,8], Joachim Reidl[1,9,10], Pedro A Sánchez-Murcia[2], Klaus Zangger[3,9,10], Tea Pavkov-Keller[1,9,10]\***

[1]Institute of Molecular Biosciences, University of Graz, Graz, Austria; [2]Laboratory of Computer-Aided Molecular Design, Division of Medicinal Chemistry, Otto-Loewi Research Center, Medical University of Graz, Graz, Austria; [3]Institute of Chemistry / Organic and Bioorganic Chemistry, Medical University of Graz, Graz, Austria; [4]Diagnostic and Research Institute of Hygiene, Microbiology and Environmental Medicine, Medical University of Graz, Graz, Austria; [5]Biological Small Angle Scattering, EMBL Hamburg, Hamburg, Germany; [6]RedShiftBio, Boxborough, United States; [7]Institute of Molecular Biology of Barcelona, Barcelona, Spain; [8]ICREA, Institució Catalana de Recerca i Estudis Avançats, Barcelona, Spain; [9]BioHealth Field of Excellence, University of Graz, Graz, Austria; [10]BioTechMed-Graz, Graz, Austria

**\*For correspondence:**
nina.gubensaek@uni-graz.at (NG);
tea.pavkov@uni-graz.at (TP-K)

**Abstract** The seventh pandemic of the diarrheal cholera disease, which began in 1960, is caused by the Gram-negative bacterium *Vibrio cholerae*. Its environmental persistence provoking recurring sudden outbreaks is enabled by *V. cholerae's* rapid adaption to changing environments involving sensory proteins like ToxR and ToxS. Located at the inner membrane, ToxR and ToxS react to environmental stimuli like bile acid, thereby inducing survival strategies for example bile resistance and virulence regulation. The presented crystal structure of the sensory domains of ToxR and ToxS in combination with multiple bile acid interaction studies, reveals that a bile binding pocket of ToxS is only properly folded upon binding to ToxR. Our data proposes an interdependent functionality between ToxR transcriptional activity and ToxS sensory function. These findings support the previously suggested link between ToxRS and VtrAC-like co-component systems. Besides VtrAC, ToxRS is now the only experimentally determined structure within this recently defined superfamily, further emphasizing its significance. In-depth analysis of the ToxRS complex reveals its remarkable conservation across various *Vibrio* species, underlining the significance of conserved residues in the ToxS barrel and the more diverse ToxR sensory domain. Unravelling the intricate mechanisms governing ToxRS's environmental sensing capabilities, provides a promising tool for disruption of this vital interaction, ultimately inhibiting *Vibrio's* survival and virulence. Our findings hold far-reaching implications for all *Vibrio* strains that rely on the ToxRS system as a shared sensory cornerstone for adapting to their surroundings.

## Editor's evaluation

This study provides important insights into the structure and mechanism of the sensory protein complex ToxR/S that is associated with the survival and virulence of the cholera pathogen. The structural studies are solid and supported by a series of biophysical experiments revealing a split, periplasmic protein binding interface for bile acid. Results are of interest to protein biochemistry and pharmacology where they may open new routes for the treatment of cholera disease.

**eLife digest** Cholera is a contagious diarrheal disease that leads to about 20,000 to 140,000 yearly deaths. It is caused by a bacterium called *Vibrio cholerae*, which can survive in harsh conditions and many environments. It often contaminates water, where it lives in an energy-conserving mode. But when humans consume *Vibrio cholerae*-contaminated water or food, the bacterium can sense its new environment and switch into a high-energy consuming state, causing fever, diarrhea, and vomiting.

*Vibrio cholerae* recognizes bile acid in the human stomach, which signals that the bacterium has reached ideal conditions for causing disease. So far, it has been unclear, how exactly the bacterium detects bile acid. Understanding how these bacteria sense bile acid, could help scientists develop new ways to prevent cholera outbreaks or treat infections.

Gubensäk et al. analysed two proteins from the *Vibrio cholerae* bacterium, called ToxR and ToxS, which are located below the bacteria's protective membrane. More detailed analyses showed that the two proteins bind together, forming a bile-binding pocket. When correctly assembled, this bile-sensing machine detects bile concentrations in the body, allowing the bacterium to adapt to the local conditions. Using crystal structures, a series of interaction studies, and modeling software, Gubensäk et al. detailed step-by-step how the two proteins sense bile acid and help the bacteria adapt and thrive in the human body.

The results confirm the results of previous studies that implicated ToxR and ToxS in bile sensing and provide new details about the process. Scientists may use this information to develop new ways to interfere with the bacteria's bile-sensing and gut adaptation processes. They may also use the information to screen for existing drugs that block bile sensing and then test as cholera treatments or prevention strategies in clinical trials. New cholera treatment or prevention approaches that don't rely on antibiotics may help public health officials respond to growing numbers of cholera outbreaks and to prevent the spread of antibiotic-resistant bacteria.

## Introduction

The Gram-negative bacterium *Vibrio cholerae* is the causative agent of the diarrheal cholera disease, which is pandemic since 1960 (*Hu et al., 2016*). Its dangerousness is highlighted by its sudden outbreaks and environmental persistence causing 21 000–143,000 000 deaths worldwide per year (*Ali et al., 2015*) including long-lasting environmental and economic damage (*Kanungo et al., 2022*).

*V. cholerae* exhibits a life cycle between dormant and virulent state, enabled by its rapid adaption to changing environments (*Almagro-Moreno et al., 2015b*; *Bari et al., 2013*). This survival mechanism is maintained via sensory proteins reacting to environmental conditions and substances (*DiRita et al., 1991*; *Hung and Mekalanos, 2005*). For entero-pathogens like *V. cholerae*, bile acid represents one of the major components for virulence activation, pressuring survival strategies (*Hung et al., 2006*; *Hung and Mekalanos, 2005*; *Li et al., 2016*; *Midgett et al., 2017*).

ToxR is a transmembrane transcription factor (*Figure 1—figure supplement 1*) involved in the regulation of numerous genes, not only virulence associated, and can function as an activator, co-activator and repressor (*Bina et al., 2003*; *Champion et al., 1997*; *Lee et al., 2000*; *Morgan et al., 2011*; *Skorupski and Taylor, 1997*; *Wang et al., 2002*; *Welch and Bartlett, 1998*). ToxR periplasmic domain is proposed to act as environmental sensor, being able to bind bile acids (*Midgett et al., 2017*; *Midgett et al., 2020*) and consequently activate transcription with its cytoplasmic DNA binding domain (*Gubensäk et al., 2021a*; *Morgan et al., 2011*; *Morgan et al., 2019*; *Pfau and Taylor, 1996*; *Withey and DiRita, 2006*), thus inducing a switch of outer membrane proteins from OmpT to OmpU (*Simonet et al., 2003*; *Wibbenmeyer et al., 2002*). Since OmpU is more efficient in excluding bile salts due to its negatively charged pore (*Duret and Delcour, 2006*; *Simonet et al., 2003*), bile-induced ToxR activation enables *V. cholerae* survival in the human gut (*Wibbenmeyer et al., 2002*).

ToxS is built of a periplasmic, a transmembrane and a short cytoplasmic region (*Figure 1—figure supplement 1*). The periplasmic domain of ToxS (ToxSp) interacts with the periplasmic domain of ToxR (ToxRp) forming a stable heterodimer (ToxRSp; *Gubensäk et al., 2021b*) thereby protecting ToxR from periplasmic proteolysis and enhancing its activity (*Almagro-Moreno et al., 2015a*; *Almagro-Moreno et al., 2015c*; *Gubensäk et al., 2021b*; *Lembke et al., 2018*; *Pennetzdorfer et al., 2019*).

The exact mechanism of ToxR functionality is not clear yet, it was proposed that ToxR binds DNA as a homodimer at the so called 'tox-boxes' which represent direct repeat DNA motifs (*Crawford et al., 1998*; *Goss et al., 2013*; *Krukonis and DiRita, 2003*; *Ottemann and Mekalanos, 1996*; *Pfau and Taylor, 1996*; *Withey and DiRita, 2006*). Recently, it was shown that ToxR uses a topological DNA recognition mechanism by recognizing DNA structural elements rather than base sequences (*Canals et al., 2023*). Also, it is suggested that by multiple binding events of ToxR to promoter regions, membrane attached transcription regulation of ToxR is enabled (*Canals et al., 2023*). The versatile functionality of ToxR suggests a general role of ToxRS in *Vibrio* strains, for example the sensing and consequent adaption to changing environmental conditions (*Chen et al., 2018*; *Provenzano et al., 2000*).

Bile significantly alters the virulence factor production. Nevertheless, the exact mechanism remains unclear. Studies showed opposite outcomes in regard of up- or downregulation of virulence factors by bile (*Bina et al., 2021*; *Gupta and Chowdhury, 1997*; *Hung and Mekalanos, 2005*; *Midgett et al., 2017*; *Xue et al., 2016*; *Yang et al., 2013*). On the one hand a bile induced reduction of virulence factor production was observed (*Gupta and Chowdhury, 1997*) via inhibiting ToxT DNA binding ability (*Plecha et al., 2015*). On the other hand bile seems to enhance TcpP and ToxR activity (*Yang et al., 2013*) and even induce ToxR ability to activate *ctx* without ToxT in classical biotype *V. cholerae* strains (*Hung and Mekalanos, 2005*). Nevertheless, direct activation of *ctx* by ToxRS by bile acids is dependent on *ctx* promotors, differing between strains (*Hung and Mekalanos, 2005*). Concluding, the effect of bile on *V. cholerae* virulence seems to be complex and probably dependent on multiple other factors for example calcium concentration and oxygen levels (*Hay et al., 2017*; *Sengupta et al., 2014*).

The presented crystal structure of sensory domains of *V. cholerae* ToxR and ToxS reveals ToxS as a main environmental sensor in *V. cholerae*. The ToxRS complex exposes a bile binding pocket inside ToxS lipocalin-like barrel that is only properly built via stabilization by newly formed structural elements of ToxR. We performed multiple interaction experiments combined with extensive molecular dynamic MD simulations, to eventually present a bile binding ToxRSp complex, revealing contributions from both proteins to the interaction with bile acid. ToxRSp shows structural and functional similarities with bile sensing complex VtrAC from *V. parahaemolyticus* (*Alnouti, 2009*; *Li et al., 2016*; *Tomchick et al., 2016*) thus supporting a common superfamily of co-component signal transduction systems whose sensory function is strictly connected to an obligate heterodimer formation (*Kinch et al., 2022*). AlphaFold-Multimer (*Evans et al., 2021*; *Jumper et al., 2021*) structure predictions furthermore reveal a conserved fold of ToxS in different *Vibrio* species, in contrast to ToxR exhibiting a structural variability among different *Vibrio* strains.

## Results

### Periplasmic domains of *V. cholerae* ToxRS form an obligate dimer

The crystal structure of ToxRp and ToxSp reveals the formation of a heterodimer, with ToxRp contributing secondary structure elements to ToxSp otherwise unstable ß-barrel fold (*Figure 1*, *Figure 2*, pdb: 8ALO). The complex forms spontaneously upon addition of both proteins.

ToxRp in the complex has an αß fold which is formed by an α-helix flanked by an anti-parallel hairpin and a three-stranded anti-parallel ß-sheet, followed by a C-terminal helix (*Figure 1*). Together, the ß-strands form a five-stranded anti-parallel ß-sheet. The C-terminal $Cys293_{ToxRp}$ is involved in a solution-oriented disulphide bond with $Cys236_{ToxRp}$ of the N-terminal helix (*Figure 3*).

ToxSp forms a lipocalin-like fold consisting of an eight-stranded ß-barrel stabilized by intermolecular H-bond network and flanked by two α-helices located at the openings of the barrel (*Figure 2*, *Supplementary file 2*). For a correct barrel formation ToxSp ß1 and ß8 would need to be close enough for building main chain interactions. However, ToxSp ß8 contacts only the first three residues of nine residues long ToxSp ß1 (*Figure 2*). ToxRp ß5 is positioned into this gap and forms main chain H-bonds with remaining ToxSp ß1 residues. Additionally, close side-chain interactions between ToxRp ß4 and ToxSp ß8 further stabilize ToxSp ß8 positioning (*Figure 2*). ToxRp ß4 and ß3 are also involved in side-chain interactions with ToxSp ß7 and ß6 completing the strong complex formation. The residue Arg281 located in ToxRp ß5 forms an intermolecular salt bridge with Asp142 of ToxSp α2, which restricts the opening of the barrel (*Figure 2*).

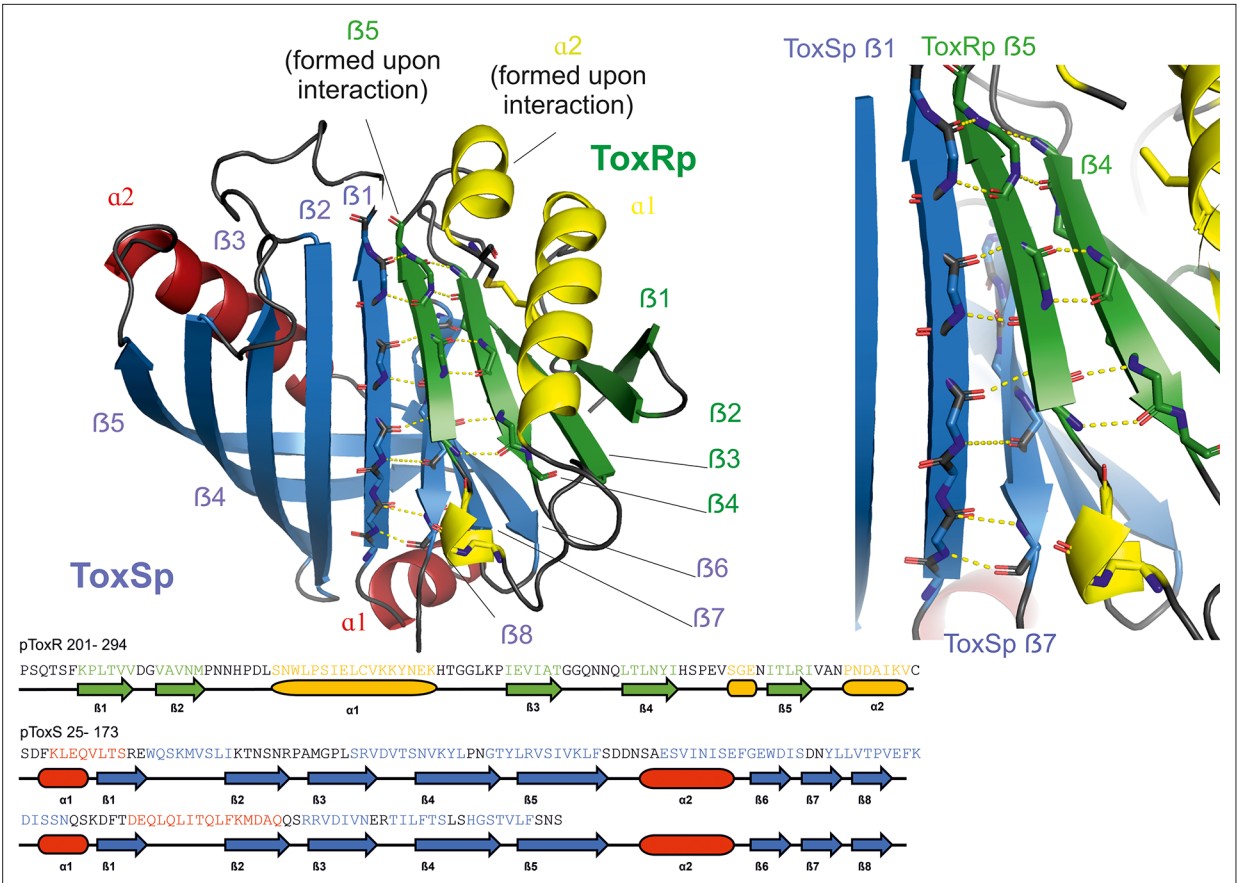

**Figure 1.** Heterodimer formation of ToxRSp (pdb: 8ALO). ToxRp ß strand 5 completes main chain hydrogen bond network of ToxSp ß-barrel formation, by interaction with ToxSp ß1. ToxSp ß8 position is stabilized by main chain hydrogen bonds with ToxSp ß1, as well as side chain interactions with ToxRp ß5 and ß4. ToxRp ß5 and α2 are formed upon ToxSp interaction. ToxRSp crystal structure was determined by molecular replacement using an AlphaFold (*Evans et al., 2021*; *Jumper et al., 2021*) model as template.

The online version of this article includes the following figure supplement(s) for figure 1:

**Figure supplement 1.** Inner membrane spanning domains of *Vibrio cholerae* regulators ToxR and ToxS.

**Figure supplement 2.** ToxRSp structure highlighting L33ToxSp core stabilizing hydrophobic network.

**Figure supplement 3.** Structural alignment of bile interacting protein complexes ToxRS *V. cholerae* and VtrAC *V. parahaemolyticus*.

DSSP secondary structure analysis (*Kabsch and Sander, 1983*; *Touw et al., 2015*) indicates a ten-stranded intermolecular ß-sheet consisting of ß1-ß5 of ToxSp and ß1-ß5 of ToxRp. ToxSp five stranded ß-sheet (ß1-ß5) and three-stranded ß-sheet (ß6-ß8), result in a ß-barrel formation, which is stabilized mainly via hydrophobic core interactions and interactions with ToxRp ß5 (*Figure 1*).

Mutational studies reveal that ToxSp mutant L33S results in increased proteolysis of ToxRp indicating a loss of function of ToxS (*Pfau and Taylor, 1998*). The crystal structure of ToxRSp shows that L33 is located at ToxSp α1 at the N-terminal membrane-oriented opening of the barrel of ToxSp (*Figure 1—figure supplement 2*). L33 forms central hydrophobic interactions with apolar residues of ToxSp ß strands stabilizing the core of the barrel (*Figure 1—figure supplement 2*.). A serine to leucine mutation probably disrupts the hydrophobic interactions and may result in a loss of structure, thus explaining the inability of ToxSp L33S mutant to protect ToxR from proteolysis (*Pfau and Taylor, 1998*).

There are currently no structures of proteins with significant sequence homology to the ToxRSp complex available in the protein data bank. However, despite the lack of sequence identity the VtrAC complex of *V. parahaemolyticus* (*Tomchick et al., 2016*; pdb code: 5KEV, 5KEW) has striking structural and functional similarities with ToxRSp from *V. cholerae* (*Figure 1—figure supplement 3*, all-atom RMSD 5.348 Å calculated for 836 atoms; sequence identity: ToxRp-VtrA: 8.1%; ToxSp-VtrC: 11.7%).

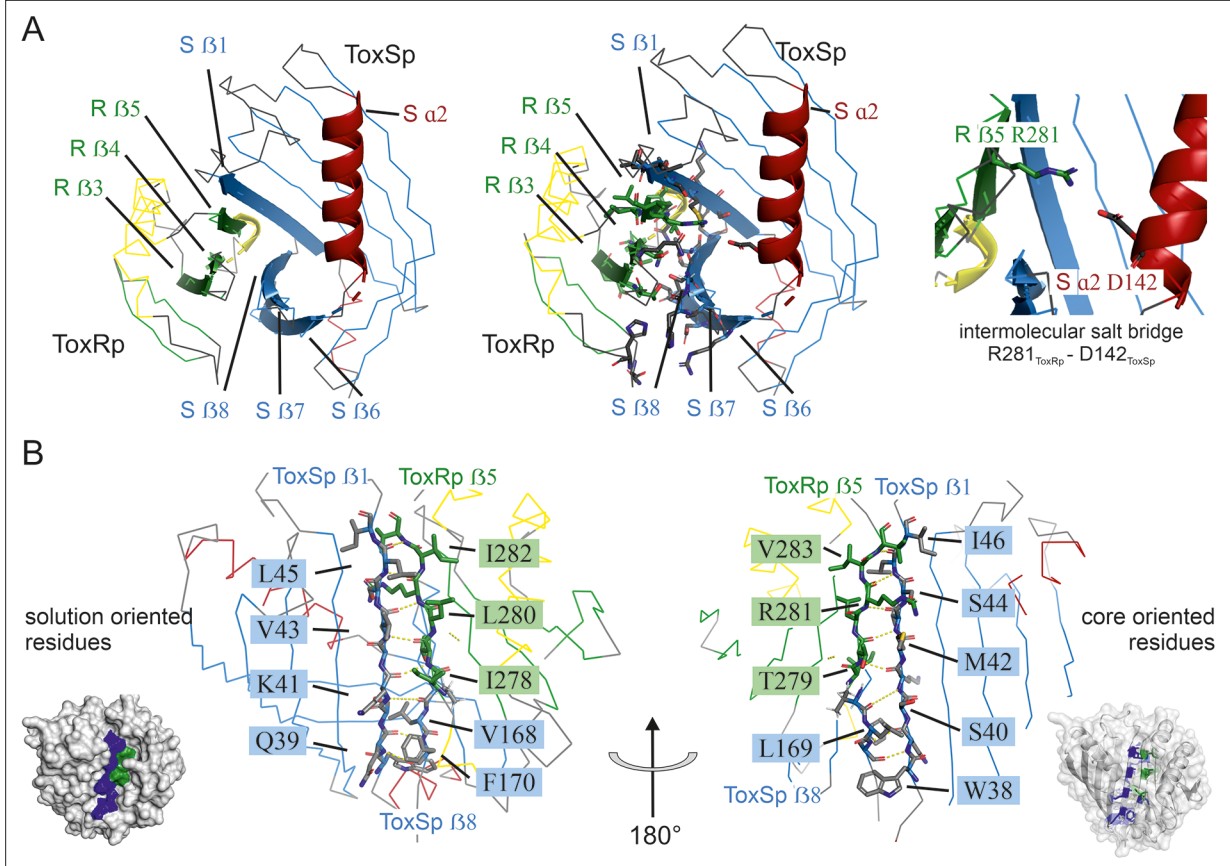

**Figure 2.** Detailed description of ToxRSp interface. (**A**) Overview of ToxRSp interface. The interaction is mainly established via ToxRp ß strands 3, 4 and 5, and ToxSp ß strands 6, 7 and 8. Additionally, an intermolecular salt bridge is formed between ToxRp R281 and ToxSp D142 located at the opening of ToxSp barrel. (**B**) The interprotein main chain H-bond pattern of ToxRSp. The interprotein H-bond network is established between ToxRp ß5 and ToxSp ß1 thereby filling the gap between ToxSp ß1 and ß8, which interact only in the C-terminal region of the strands.

Similar to ToxRS, the bile sensing functionality of VtrAC induces an essential step for virulence activation upon human gastro-intestinal infection. VtrAC activates the production of main virulence factor VtrB which enables the production of a type III secretion system 2 (T3SS2) for the injection of virulence factors into host cells (*Kaval et al., 2023*; *Zou et al., 2023*).

### Intrinsically disordered ToxRp C-terminus folds into structural elements essential for the complex formation with ToxSp

Small-angle X-ray scattering (SAXS) experiments indicate a dimer formation of ToxSp in the absence of ToxRp (*Figure 6—figure supplement 1*, *Figure 6—figure supplement 2*, *Supplementary file 1c*) supporting the recently proposed HDOCK ToxS-ToxS dimer (*Canals et al., 2023*). Nevertheless, aggregation of ToxSp in solution occurs rapidly suggesting an unstable dimer formation (*Gubensäk et al., 2021b*; *Figure 6—figure supplement 3*). The instability of ToxSp can be explained by the inability of the first and the last ß strand of the barrel to form strong hydrogen bonding, thus resulting in an unstable hydrophobic core and consequently aggregation. Interaction with ToxRp likely protects otherwise exposed hydrophobic regions of ToxSp (*Figure 1*, *Figure 2*).

Comparison of the ToxRSp crystal structure with our previously determined NMR structure of ToxRp (pdb: 7NN6) (*Gubensäk et al., 2021b*), shows the formation of additional secondary structure elements upon complex formation (*Figure 3*). Unbound ToxRp has a long intrinsically disordered C-terminus in solution, containing the second cysteine Cys293$_{ToxRp}$ forming a disulphide bond with Cys236$_{ToxRp}$ located in the middle of helix 1 (*Gubensäk et al., 2021b*). When bound to ToxSp, the unstructured region of the C-terminus of ToxRp performs a disorder to order transition, by forming two additional structural elements: short α helix 2 and ß strand 5 (*Figure 3*). The newly formed short α

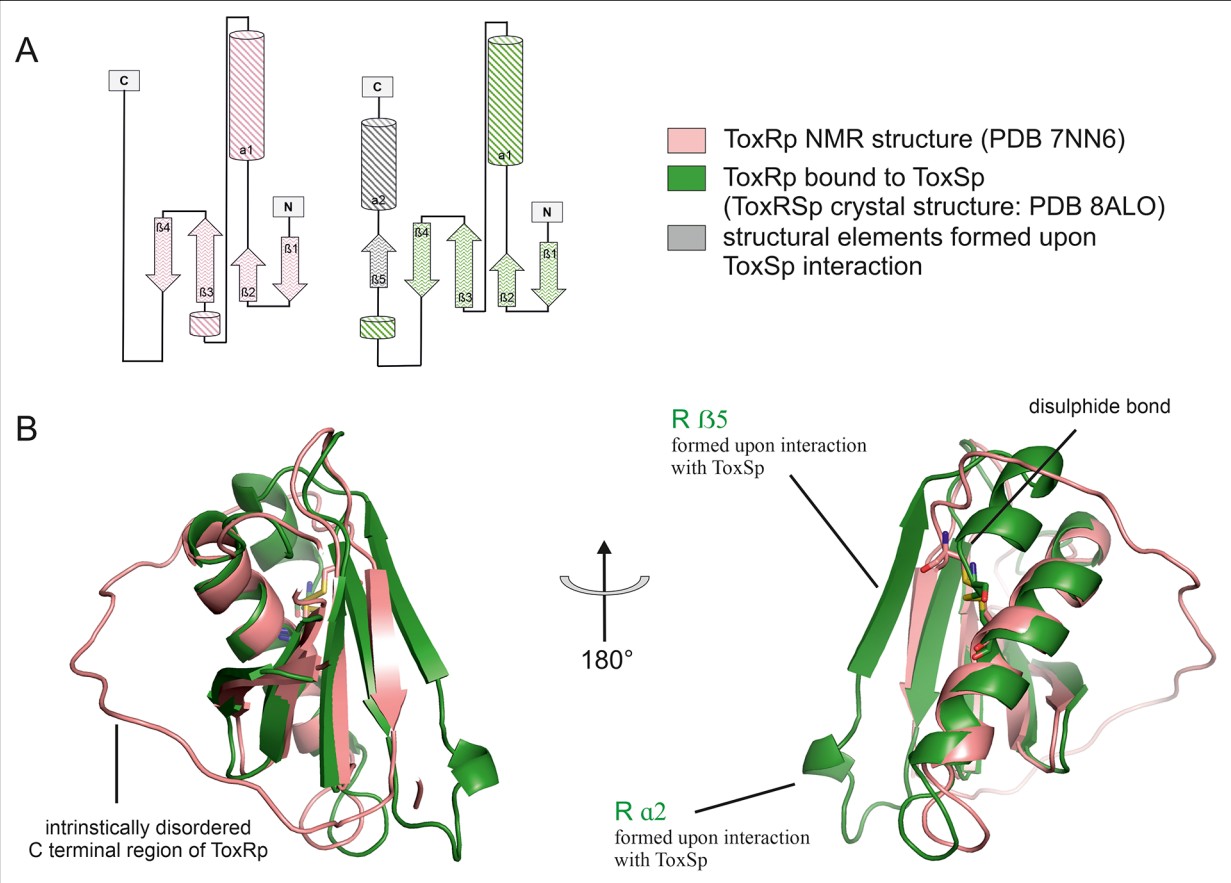

**Figure 3.** Comparison of ToxRp NMR structure (salmon) to ToxRp bound to ToxSp (green). Upon binding to ToxSp, ToxRp intrinsically disordered C-terminal region forms ß5 and α2. (**A**) Topology diagram of ToxRp and ToxRp bound to ToxSp. Newly formed structural elements are highlighted in grey. (**B**) Structural alignment between ToxRp (PDB 7NN6) and ToxRp in complex with ToxSp (PDB 8ALO). The C-terminal disulphide bond is shown in sticks.

helix 2 contacts the turn between ToxSp ß1 and ß2, as well as residues Leu45$_{ToxSp}$ and Ile46$_{ToxSp}$ located at the C-terminal part of ToxSp ß1 thus increasing the interaction interface and further stabilizing the fold. Although the C-terminal region undergoes extreme structural changes upon complex formation, the orientation of the disulphide bond does not change drastically.

ToxRp ß strand 5 is only formed upon complex formation with ToxSp and forms the essential basis for the interaction by building a stabilizing hydrogen bonding network with ß strands 1 and 8 of ToxSp (**Figure 2**). Newly formed ToxRp ß5 also forms hydrogen bonds with ToxRp ß4. In unbound ToxRp, polar residues of ß4 are pointing towards solution, whereas hydrophobic parts contribute to the hydrophobic core. Nevertheless, ToxRp ß4 does not reveal significant conformational changes upon ToxRSp complex formation.

## ToxSp in complex with ToxRp contains a bile binding pocket

ToxRSp bile interaction was tested using NMR (**Becker et al., 2018**) (saturation transfer difference STD **Figure 6—figure supplement 3**, **Figure 6—figure supplement 4**), chemical shift perturbation CSP (Figure 7, **Figure 6—figure supplement 5**) and diffusion ordered spectroscopy DOSY (Figure 6B, **Figure 6—figure supplement 6**), as well microfluidic modulation spectroscopy MMS (**Figure 4**, **Figure 4—figure supplement 1**), size exclusion chromatography (Figure 6C) and SAXS (Figure 6A, **Figure 6—figure supplement 1**, **Figure 6—figure supplement 2**, **Supplementary file 1c**). Interaction experiments were performed with bile acid sodium cholate hydrate S-CH. All mentioned experiments confirm an interaction of the ToxRSp complex with bile.

The lipocalin-like fold of ToxSp forms a hydrophobic cavity (**Figure 5**, **Figure 5—figure supplement 1**), similar to the bile binding pocket of VtrC in complex with VtrA from *V. parahaemolyticus* (**Tomchick et al., 2016**). Using extensive MD simulations enabled a detailed analysis of the

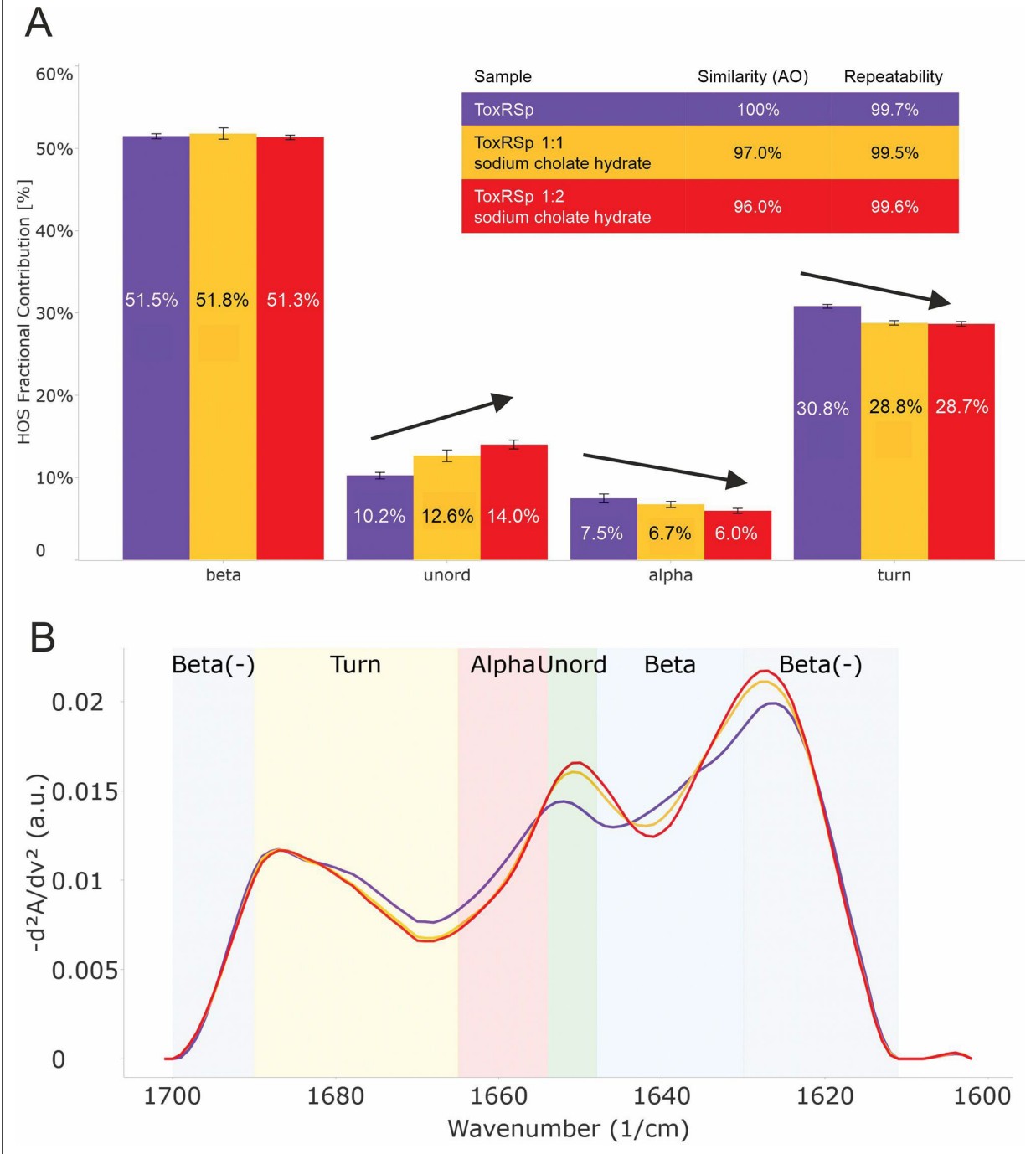

**Figure 4.** Analysis of microfluidic modulation spectroscopy MMS experiments. MMS experiments of ToxRSp with different S-CH concentrations show conformational changes upon binding of bile to ToxRSp. Additional structural rearrangements could be detected upon higher additions of bile. (**A**) Bar chart of the relative abundance of secondary structural motifs based on Gaussian deconvolution of the corresponding spectra. (**B**) MMS spectra (baselined, inverted 2nd Derivative) of ToxRSp with and without bile, showing spectral changes upon higher bile additions.

The online version of this article includes the following source data and figure supplement(s) for figure 4:

**Source data 1.** MMS data.

**Figure supplement 1.** MMS titration experiment with ToxRSp and S-CH.

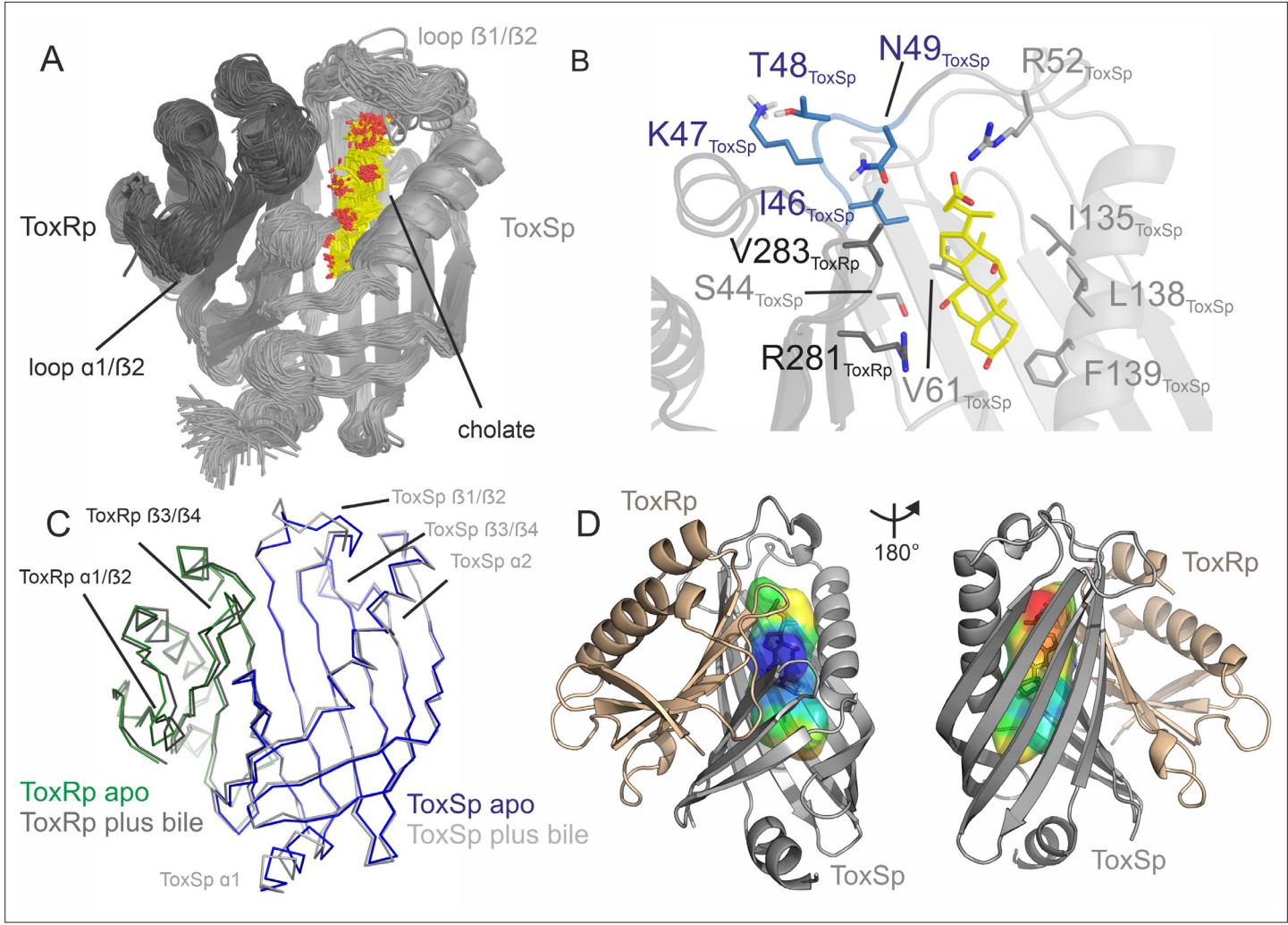

**Figure 5.** ToxRSp binding pocket for bile acid. (**A**) Superimposition of 100 structures of the ToxRSp complex (dark grey/light grey cartoons) with cholate (sticks, C-atoms in yellow) along one representative classical MD simulation. (**B**) Detailed view of the binding mode of cholate to ToxRSp. The residues with the larger contribution to the binding of the ligand are highlighted as well as residues 46–49 of ToxSp located on loop β1/β2. (**C**) Overlay of ToxRSp apo and bile-bound structures. Conformational changes mainly occur in ToxSp loop β1/β2 and ToxRp loop α1/β2. ToxSp helices α1 and α2 and ToxRp loop β3/β4, both close to the ToxRSp interface, experience minor conformational rearrangements upon bile interaction. (**D**) Hydrophobic properties of the bile binding cavity calculated with the CavMan (v. 0.1, 2019, Innophore GmbH plugin in PyMOL). The entrance of the cavity (blue to green) faces the solvent exhibiting a hydrophilic environment, in contrast to the buried hydrophobic areas (yellow to red). For the analysis of the hydrophobicity of the cavities the program VASCo (*Steinkellner et al., 2009*) was used; cavities were calculated using a LIGSITE algorithm (*Hendlich et al., 1997*).

The online version of this article includes the following source data and figure supplement(s) for figure 5:

**Source data 1.** ToxRSp MD data.

**Figure supplement 1.** Hydrophobic cleft of ToxSp binding pocket in complex with ToxRp. Hydrophobic (ochre) and hydrophilic (teal) surface representation of ToxS calculated with ChimeraX mlp. The bile binding cavity is highly hydrophobic (arrow).

protein-ligand interface. Upon binding of bile acid mainly loop regions undergo major conformational changes (*Figure 4*, *Figure 5*). Comparison of ToxRSp apo and bile-bound state reveals that ToxSp β1/β2 loop and ToxRp α1/β2 loop experience strongest rearrangements, whereas loop β3/β4$_{ToxSp}$, α1$_{ToxSp}$, α2$_{ToxSp}$ and loop β3/β4$_{ToxRp}$ reveal only minor structural changes (*Figure 5C*). Similarly, MMS spectra show significant spectral changes in disordered, turn and helical regions (*Figure 4*).

The nature of the interactions of bile acid with the protein complex are mostly hydrophobic. The carboxylic moiety forms a strong interaction with Arg52$_{ToxSp}$. Additionally, cholate interacts with residues Ser44$_{ToxSp}$, Ile135$_{ToxSp}$, Leu138$_{ToxSp}$, Phe139$_{ToxSp}$, Val283$_{ToxRp}$, and Arg281$_{ToxRp}$ (*Figure 5*). Both

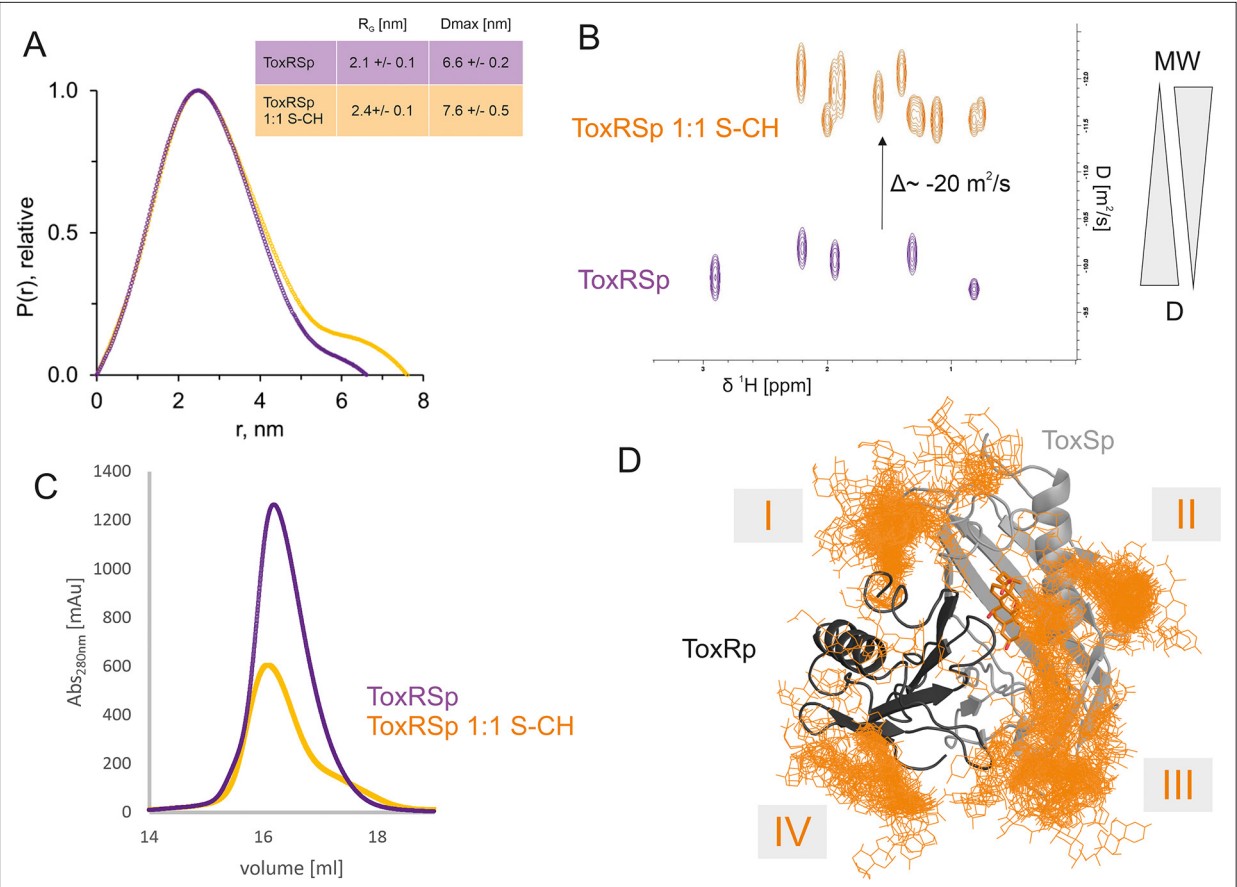

**Figure 6.** ToxRSp bile interaction experiments propose additional bile interaction areas I-IV. (**A**) SEC-SAXS experiments with ToxRSp and S-CH reveal an increase of the radius of gyration $R_G$ and the effective maximum particle dimension Dmax upon bile addition. (**B**) Superimposed NMR DOSY spectra of ToxRSp with and without bile acid S-CH. Addition of bile acid causes a decrease of the diffusion coefficient due to an increase of molecular weight. (**C**) SEC experiments with ToxRSp and S-CH reveal a slight decrease of retention volume upon bile acid presence from 16.19 ml to to 16.08 ml and a broadening of the peak. (**D**) MD determined surface exposed bile interacting regions of ToxRSp. Besides the bile binding cavity of ToxSp, four regions (marked as I-IV) on the ToxRSp surface could be mapped as additional bile binding areas. Region I and III involves both proteins, whereas region II is located on ToxSp and region IV on ToxRp. The representative structure of the ToxRSp complex (ToxRp: dark grey, ToxSp: light grey) is shown with cholate bound at the cavity (orange) and additional cholate molecules attach to surface exposed regions of ToxRSp.

The online version of this article includes the following source data and figure supplement(s) for figure 6:

**Source data 1.** SEC data.

**Figure supplement 1.** Analysis of size-exclusion chromatography coupled with solution small-angle X-ray scattering SEC-SAXS experiments.

**Figure supplement 2.** SAXS-derived chromatogram showing Integrated intensities (0.1 nm−1<s < 0.8 nm−1) vs. elution of indicated samples.

**Figure supplement 3.** Saturation transfer difference STD NMR spectra of ToxRSp with S-CH.

**Figure supplement 3—source data 1.** STD ToxRSp plus S-CH.

**Figure supplement 4.** Saturation transfer difference NMR spectra of ToxSp with sodium cholate hydrate.

**Figure supplement 4—source data 1.** STD ToxSp plus S-CH.

**Figure supplement 5.** $^1$H NMR spectrum of ToxRSp upon S-CH addition.

**Figure supplement 6.** Superimposed NMR DOSY spectra of ToxRSp with and without bile acid sodium cholate hydrate.

**Figure supplement 7.** SEC runs of ToxRSp with and without bile acid S-CH.

**Figure supplement 8.** NMR interaction experiment of ToxRSp with cyclo-PhePro (cFP).

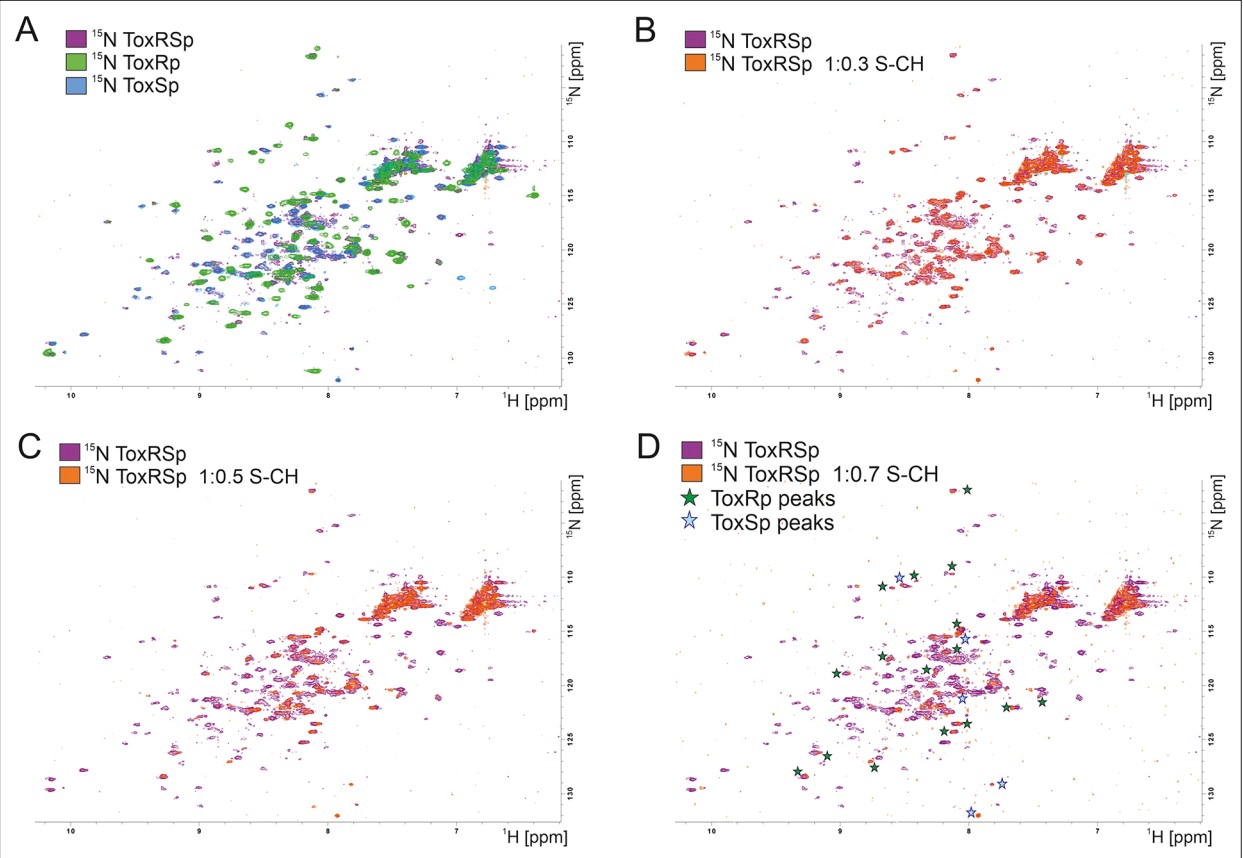

**Figure 7.** NMR titration of S-CH to ¹⁵N labelled ToxRSp. (**A**) Overlay of ¹⁵N-HSQC spectra of ¹⁵N-labelled ToxRSp (purple) with ¹⁵N-labelled ToxRp (green) and ToxSp (blue) for ascription of peaks to each of the proteins. (**B–D**) Overlay of ¹⁵N-HSQC spectra of ¹⁵N-labelled ToxRSp before (purple) and after the addition of increasing ratios of S-CH (orange): 0.3 (**B**), 0.5 (**C**), 0.7 (**D**). Upon bile acid addition signal intensities significantly decrease. Upon higher additions of bile acid mainly ToxRp peaks are visible indicated by green asterisk in (**D**), whereas most ToxSp signals disappear indicating direct interaction with S-CH. Remaining ToxSp signals are marked with blue asterisk.

The online version of this article includes the following source data for figure 7:

**Source data 1.** NMR titration 1 ToxRSp plus S-CH.

**Source data 2.** NMR titration 2 ToxRSp plus S-CH.

proteins contribute to the bile acid interaction, although ToxSp residues are forming a major part of the interface.

## ToxRSp has additional surface exposed bile binding patches

ToxRSp interaction experiments propose an increase of molecular weight upon bile addition as shown in *Figure 6*. SEC-SAXS experiments indicate an increase of molecular weight, radius of gyration and effective maximum particle dimension of ToxRSp upon addition of equimolar amounts of bile acid (*Figure 6A*, *Figure 6—figure supplement 1*, *Figure 6—figure supplement 2*, *Supplementary file 1c*). Similar outcomes are observed by SEC runs of ToxRSp with and without bile acid using a Superdex200 column resulting in a broadening of ToxRSp-bile acid peak and slightly decreased retention times (*Figure 6C*, *Figure 6—figure supplement 7*). NMR DOSY experiments of ToxRSp bound to bile acid clearly show a decrease of the diffusion coefficient, which is linked to an increase of size and weight (*Figure 6B*, *Figure 6—figure supplement 5*). Via MMS experiments conformational changes could be detected upon bile acid binding but clearly indicate that no severe structural rearrangements for example aggregation events occur (*Figure 4*, *Figure 4—figure supplement 1*).

Due to a loss of signals upon equimolar additions of bile acid caused by the previously mentioned increase of molecular weight, NMR titration experiments were performed using small additions of

S-CH (*Figure 7*). Spectra were recorded with ToxRp and ToxSp both $^{15}$N isotopically labelled. An overlay of spectra of only ToxRp or ToxSp $^{15}$N labelled enabled the ascription of signals to each of the proteins (*Figure 7A*). Even small additions of bile (0.1 molar ratio) to ToxRSp results in a significant decrease of signal intensity. ToxSp seems to be mostly affected by the addition of bile acid, causing a disappearance of ToxSp signals indicating direct interaction events (*Figure 7B–D*). ToxRp signals are also affected by decreased signal intensities but to a lower extent when compared to ToxSp. Upon bile additions higher than a molar ratio of 0.6, mostly signals from ToxRp are visible (*Figure 7D*). In general, all ToxRSp peaks experience decreased signal intensities caused by fast T$_2$ relaxation.

Interestingly, MMS experiments show additional structural changes of ToxRSp upon higher bile acid concentrations, with maximum changes achieved at a double molar excess of bile acid suggesting the presence of additional binding sites on ToxRSp (*Figure 4*, *Figure 4—figure supplement 1*). Prompted by this evidences, further MD simulations of the complex ToxRSp were performed with one cholate molecule at the binding cavity in the presence of 14 additional cholate molecules randomly positioned in the solvent (*Figure 6D*). Four solvent exposed regions of the protein, which show a transient binding of cholate molecules could be identified (*Figure 6D*). Three are located on ToxSp, of which two implicate interactions with ToxRp. One interaction site is exclusively located on ToxRp. These results support the experimental finding of an increase of the molecular weight by binding of more than one cholate molecule.

The active state of ToxR is proposed to be dimeric due to its binding to direct repeat DNA sequences (*Canals et al., 2023*; *Gubensäk et al., 2021a*; *Krukonis and DiRita, 2003*; *Withey and DiRita, 2006*). Nevertheless, the increase of molecular weight observed in the interaction experiments is significantly lower than 30 kDa (*Figure 6A–C*), which corresponds to the molecular weight of ToxRSp. Interaction experiments therefore do not implement a bile induced dimerization or oligomerization event. A bile induced formation of trimers involving two ToxRp and one ToxSp or two ToxSp and one ToxRp molecules also seems unlikely. A described trimer formation would involve the disruption of ToxRSp heterodimer complexes, which exhibits a dissociation constant at nanomolar range (*Gubensäk et al., 2021b*), and includes aggregation of unstable ToxSp. Also, MMS experiments reveal subtle conformational changes of ToxRSp upon bile interaction and do not support extreme structural rearrangements. Instead of major structural changes like dimerization or loss of structure due to aggregation, the experimental results more likely indicate that the presence of bile leads to local conformational changes near the binding cavity of the ToxRSp barrel, similar to the VtrAC complex of *V. parahaemolyticus* (*Li et al., 2016*; *Tomchick et al., 2016*).

Taken together, we propose that ToxRSp bile sensing functions in a complex manner involving a binding site in ToxRSp barrel as well as surface exposed binding regions for bile acid. Bile acid interaction results in concentration dependent conformational changes of ToxRSp which may be related to ToxR transcriptional activity changes (*Eichinger et al., 2011*; *Gubensäk et al., 2021a*; *Kenney, 2002*; *Martínez-Hackert and Stock, 1997a*; *Martínez-Hackert and Stock, 1997b*).

## ToxRp ß-sheet forms a low-affinity binding region for bile

The MD determined ToxRp binding region to bile (*Figure 6D*) is also confirmed experimentally via chemical shift mapping (*Figure 8*, *Figure 8—figure supplement 1*; *Becker et al., 2018*; *Williamson, 2013*). The bile binding area is located at the solution-oriented region of the ß-sheet of ToxRp. The determined dissociation constant of 2.6±1.42 mM (*Figure 8*, *Supplementary file 1d*) exhibits a weak binding affinity of ToxRp to bile but physiologically still relevant (*Qin and Gronenborn, 2014*; *Sukenik et al., 2017*) since bile acid concentrations in the small intestine vary between 2 and 10 mM (*Hamilton et al., 2007*). Therefore, we propose that ToxRp interaction with bile acid is only relevant upon high bile concentrations and could be linked to an additional bile sensing mechanism not connected to ToxRSp complex formation.

Interaction experiments with ToxSp alone and S-CH did not reveal any binding event (*Figure 6—figure supplement 4*). Due to ToxSp structural instability (*Gubensäk et al., 2021b*) we propose that without stabilization of ToxRp, the binding pocket is not properly formed and therefore binding to bile cannot occur. Additionally, the presented bile binding complex of ToxRSp shows that residues from ToxRp are also involved in the bile interaction (*Figure 5*).

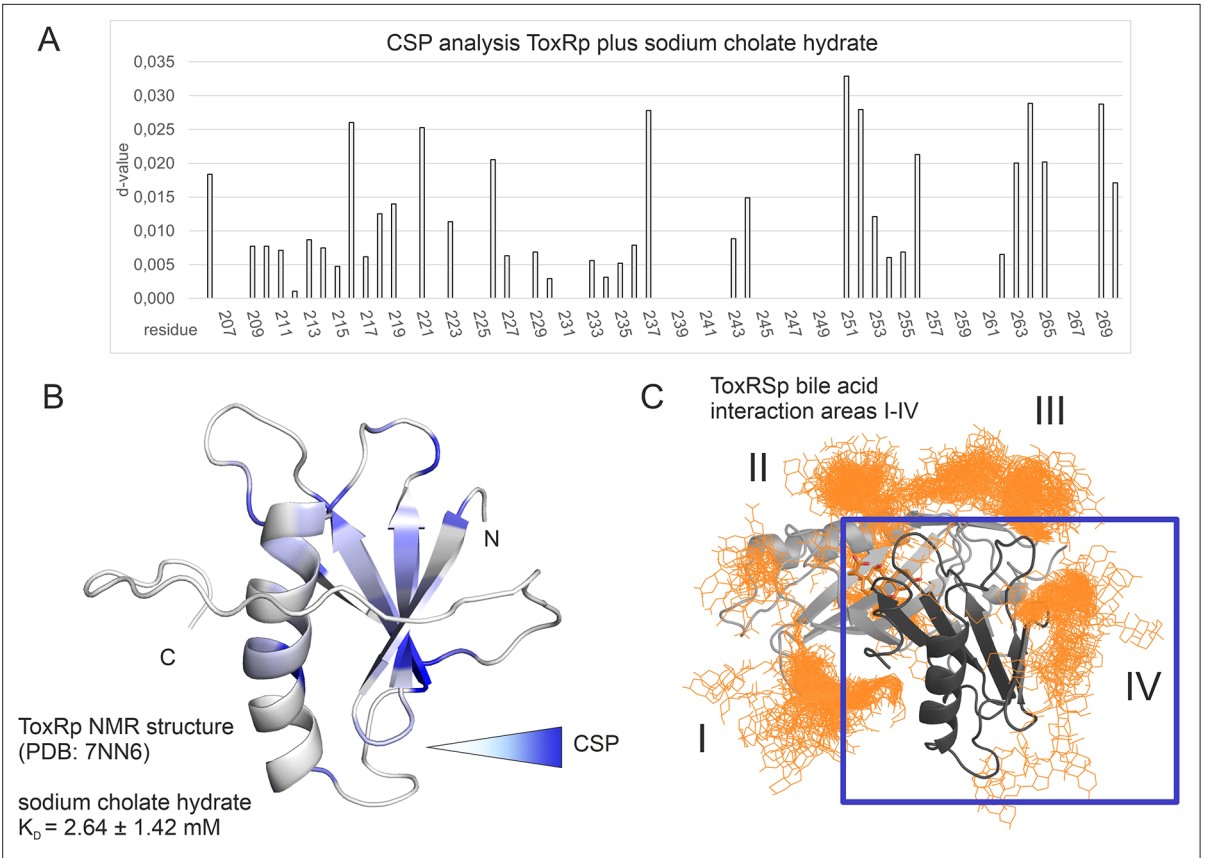

**Figure 8.** Binding studies with ToxRp and bile. (**A**) Chemical shift perturbation experiments expose residues mostly affected upon binding of S-CH. (**B**) Residues which experience a change of their chemical shift upon bile addition are coloured according to a gradient from white to blue. Residues highlighted in dark blue are mostly affected upon bile addition and are therefore most likely located at the interaction area. The calculated dissociation constant of 2.6 mM suggests a low affinity binding of bile to ToxRp. (**C**) Binding area IV determined by MD simulations (*Figure 6*) is located at the ß-sheet of ToxRp. Taken together, MD simulation and CSP experiments reveal a bile interacting area located at the ß-sheet of ToxRp proposing a bile sensing function of ToxRp independent on ToxRSp complex formation relevant only at high bile salt concentrations.

The online version of this article includes the following source data and figure supplement(s) for figure 8:

**Source data 1.** NMR titration ToxRp plus S-CH.

**Figure supplement 1.** NMR titration experiment with ToxRp and S-CH.

## Discussion

### Bile-induced ToxRS activation enables *V. cholerae* bile resistance

Sensory proteins represent an indispensable tool for *V. cholerae* to react to changes in its environment and consequently survive in harsh habitats like the human gut (*Almagro-Moreno et al., 2015b*). The interaction of inner membrane proteins ToxR and ToxS initiates a sensing function followed by signal transmission thereby causing immediate changes of the expression system of the bacterium (*Bina et al., 2003*; *Childers and Klose, 2007*; *DiRita et al., 1991*). In order to achieve bile resistance, ToxRS bile induced activation leads to a change of outer membrane protein production from OmpT to OmpU (*Morgan et al., 2011*; *Provenzano and Klose, 2000*; *Simonet et al., 2003*).

Our studies reveal a crucial heterodimer formation of the periplasmic sensory domains of *V. cholerae* thereby shaping a bile binding pocket which is only properly folded upon the complex formation. Interaction experiments show that correct heterodimer assembly of ToxRSp is critical for efficient bile sensing functionality. In vivo these findings suggest that bile-induced virulence regulation and OmpU production occurs only when ToxR and ToxS are both present for forming the obligate heterodimer, thus providing a regulatory restriction in the regulation process by bile. Although a single protein may be more efficient in sensory regulation, a co-component system offers a more strict regulation enabling fine-tuning of transcription levels according to small changes of the pathogen's environment.

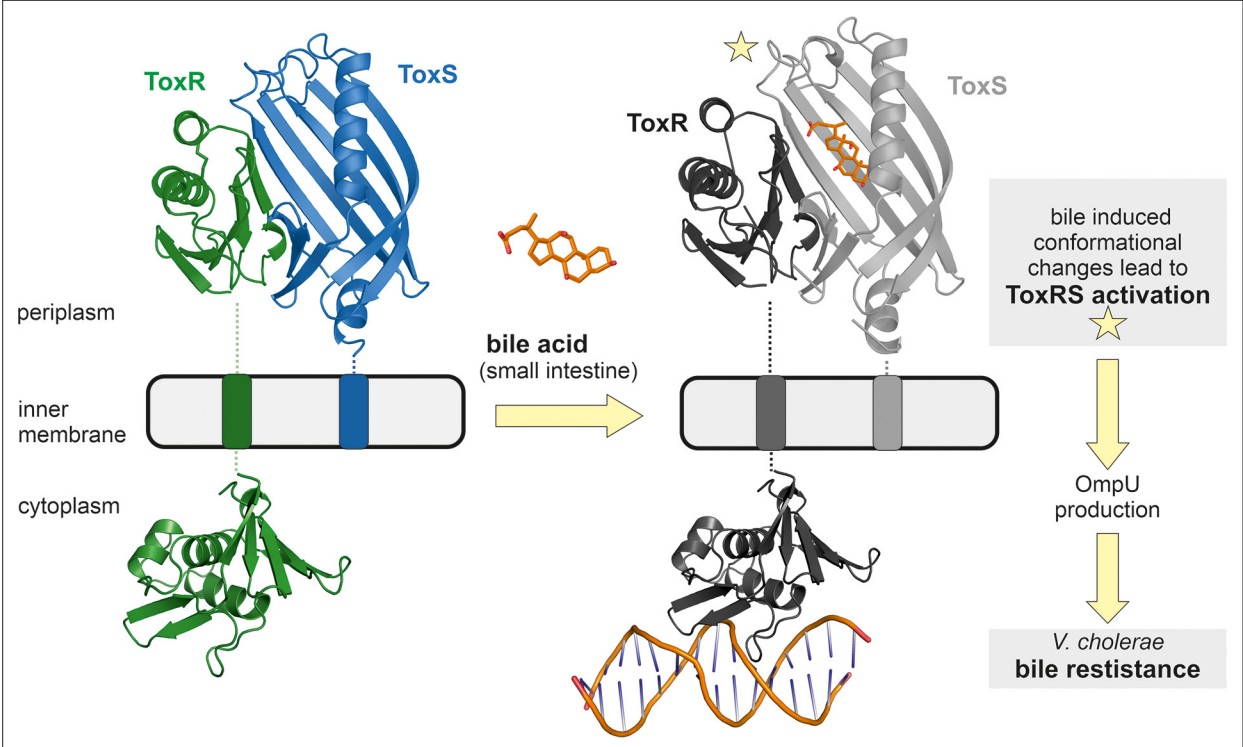

**Figure 9.** Model of bile induced activation of *V. cholerae* ToxRS. When entering the human gut *V. cholerae* senses the presence of bile acids by binding of bile to the periplasmic domains of inner membrane proteins ToxR and ToxS. The bile binding pocket is formed by ToxS and stabilized by ToxR. Subsequently, the interaction with bile induces ToxR activation which leads to a change of outer-membrane proteins from OmpT to OmpU. OmpU then provides bile resistance and thus enables the bacterium's survival in the human body. PDB accession codes: ToxRSp crystal structure: 8ALO, cytoplasmic domain of ToxR: 7NMB. The cytoplasmic domain of ToxR bound to *ompU* operon is a NMR guided HADDOCK model (*Gubensäk et al., 2021a*).

A distinctive feature of members of the 'ToxR-like' transcription factor family is the transduction of signals through the membrane without chemical modification, probably via conformational changes (*Eichinger et al., 2011*; *Gubensäk et al., 2021a*; *Kenney, 2002*; *Martínez-Hackert and Stock, 1997a*; *Martínez-Hackert and Stock, 1997b*). Thus, we suggest that the observed structural changes of the periplasmic domains of ToxRS upon bile recognition are passed on through the ToxR transmembrane domain to its cytoplasmic effector domain, thereby enhancing ToxR binding to recognition sequences and subsequently induction of transcription (*Figure 9*).

Higher additions of bile acid induce additional structural changes of ToxRSp as shown by MMS experiments (*Figure 4*). A bile acid concentration dependent mechanism could be an efficient tool for *V. cholerae* for sensing its preferred location of infection in the small intestine (SI). Bile acid concentrations also vary within the SI: postprandial bile acid concentrations are around 10 mM in the initial region of the SI, whereas the distal part contains 2 mM bile acids. (*Hamilton et al., 2007*). Still, the exact localization of *V. cholerae* within the SI is not clear (*Millet et al., 2014*). Previous studies revealed higher numbers of colony forming units (cfu) of the pathogen in the middle and distal regions of the SI (*Barbieri et al., 1999*). Nevertheless, *V. cholerae* colonization seems to depend on its motility and host factors e.g. mucins (*Millet et al., 2014*).

The presented experiments reveal that bile sensing of ToxRS functions according to a complex scheme dependent on the concentration of bile acids. ToxRSp interaction involves a binding site of ToxRSp barrel hosting one bile molecule and multiple surface exposed binding regions offering attachment sites for bile acid molecules. Bile interaction with ToxRSp induces concentration dependent conformational changes of ToxRSp probably influencing ToxR transcriptional activity changes (*Eichinger et al., 2011*; *Gubensäk et al., 2021a*; *Kenney, 2002*; *Martínez-Hackert and Stock, 1997a*; *Martínez-Hackert and Stock, 1997b*). The low affinity binding region of ToxR alone could be connected to another bile sensing mechanism, which is relevant at high concentrations of bile acid and independent on ToxRSp complex formation.

The expression system of *V. cholerae* is complex and sensitively regulated involving numerous factors and depending on environmental conditions and signals. The high-energy consuming virulent state of the pathogen is only switched on when its localization for colonization is reached. Sensing different bile acid concentrations could support the determination of the exact position of the bacterium in the human body and thereby adapting its gene expression according to environmental conditions. Bile resistant associated genes as well as factors essential for colonization are switched on at early stages in order to guarantee the survival of the bacteria, whereas other factors like the cholera toxin are produced later, enabling the spread of the bacteria and causing the main symptoms of the disease (*Almagro-Moreno et al., 2015b*; *Srivastava et al., 1980*).

The presented experiments do not indicate a bile-induced dimer- or oligomerization of the ToxRS complex, which is currently proposed as the active conformation of ToxR. Nevertheless, since experiments are performed with periplasmic domains only, a different behaviour of full-length proteins in their natural environment cannot be ruled out. Dimer- and oligomerization events may be dependent on the presence of DNA as suggested recently (*Canals et al., 2023*), and remain to be elucidated.

The strong interaction of periplasmic domains of ToxR and ToxS indicates that complex formation also occurs spontaneously in-vivo, and independent on the presence of ligands. Presented experiments were performed with already established ToxRSp complexes, to which bile acid was added after complex formation. Nevertheless, we cannot rule out that a folding-on-binding interaction of ToxRS and bile acid happens in-vivo, which remains to be elucidated.

Taken together we propose that in regard to *V. cholerae's* virulence expression system 'ToxR-regulon' ToxR transcriptional activity is guided by ToxS sensory function. The heterodimer formation is therefore inevitable for the individual functionality of each protein forming a co-dependent system. Understanding the mechanistic details of the ToxRS complex provides a relevant basis for disruption of the crucial interaction interface and consequently inhibition of *Vibrio's* adaption to its host and its virulent action.

## ToxRS belongs to the superfamily of VtrAC-like co-component signal transduction systems

*Vibrio* pathogens have a complex regulatory mechanism that allows them to survive and cause disease. ToxRS and VtrAC (*Tomchick et al., 2016*) are both *Vibrio* protein complexes fulfilling an essential function since they induce virulence by sensing bile via direct interaction (*Bina et al., 2021*; *Hay et al., 2017*; *Hung and Mekalanos, 2005*; *Tomchick et al., 2016*). Bile sensing is inevitable for the survival of gastro-entero pathogens (*Gunn, 2000*; *Tomchick et al., 2016*). On one hand outer membrane proteins need to be adapted to bile stress in order to guarantee the survival of the bacteria (*Sistrunk et al., 2016*). On the other hand, the presence of bile signals the environment of the host and therefore the virulence activating state needs to be induced in order to effectively colonize the gut.

Structurally, the protein complexes share the ß-barrel lipocalin-like formation of ToxSp/VtrC completed by a ß strand of ToxRp/VtrA, together forming a bile binding pocket (*Figure 1—figure supplement 3*). Similar to ToxSp, VtrC is not stable without its interaction partner and tends to aggregate easily (*Tomchick et al., 2016*). There is no structure of unbound VtrA available yet, so we cannot compare if VtrA also forms new structural elements upon complex formation like ToxRp (pdb code: 7NN6). Another difference between the complexes is that VtrA does not contain cysteine residues like ToxRp. The disulphide linkage represents another option for regulation in *V. cholerae*. Also, bile interaction occurs in a different manner. Whereas in ToxRSp both proteins contribute to the direct interaction with bile, in VtrAC bile binding strictly involves only VtrC (*Li et al., 2016*; *Tomchick et al., 2016*; *Zou et al., 2023*). If VtrC, in contrast to ToxS, is capable of binding bile acid on its own remains to be elucidated.

Recently, *V. cholerae* ToxRS was proposed as a distant homolog to VtrAC, forming together a superfamily of co-component signalling systems which are not linked to sequence homology (*Kinch et al., 2022*). Bacterial two-component systems typically enable signal transduction via chemical modification for example phosphorylation of a response regulator (*Mitrophanov and Groisman, 2008*). In the case of VtrAC and ToxRS signal transduction is achieved via binding of signalling molecules like bile acid, thereby inducing conformational changes which likely facilitate transcription regulation (*Eichinger et al., 2011*; *Gubensäk et al., 2021a*; *Kenney, 2002*; *Li et al., 2016*; *Martínez-Hackert and Stock, 1997a*; *Martínez-Hackert and Stock, 1997b*; *Tomchick et al., 2016*).

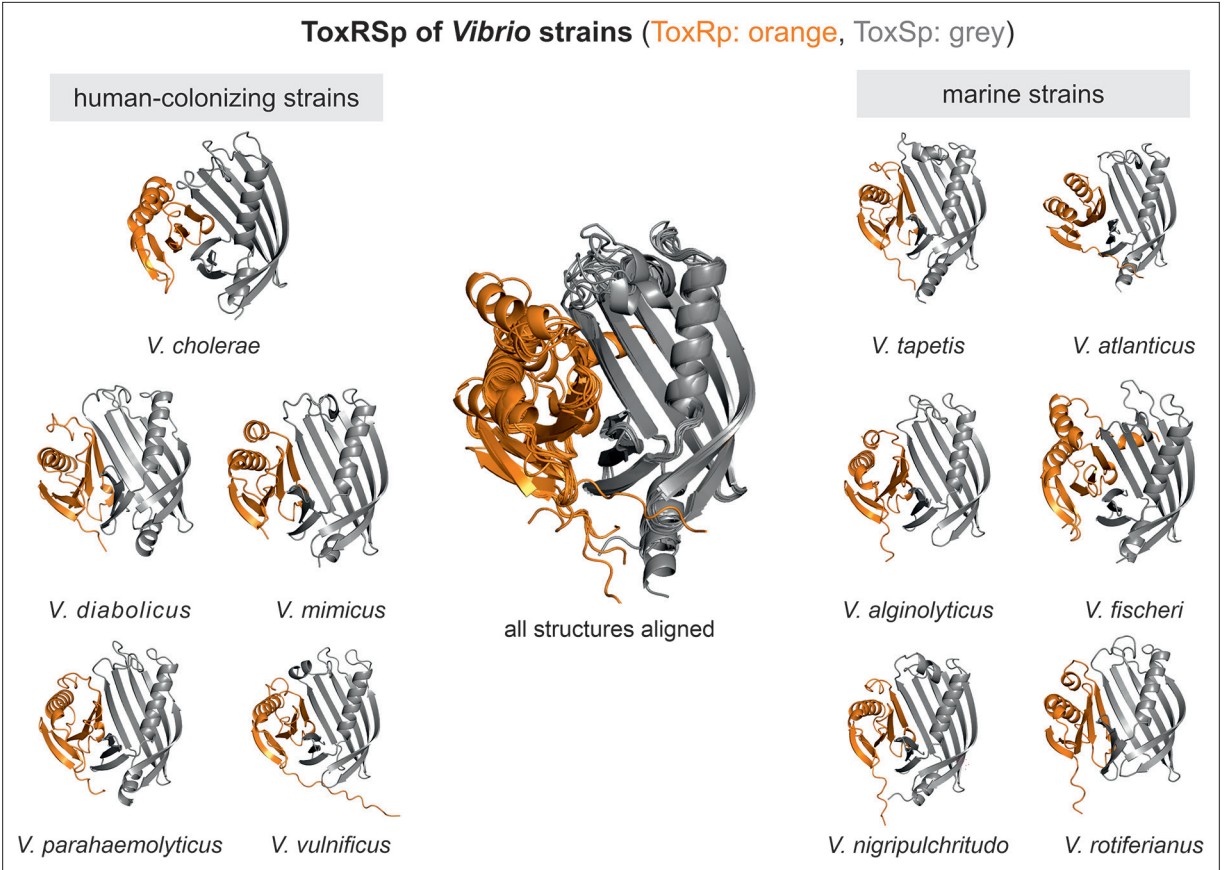

**Figure 10.** Heterodimer structures of periplasmic sensory domains of ToxRS from *Vibrio* organisms calculated using AlphaFold-Multimer (***Evans et al., 2021***; ***Jumper et al., 2021***). ToxRSp dimer formation occurs in different *Vibrio* strains. The structural alignment of all models reveal that the ToxSp fold is highly similar in whereas ToxRp seems to be more diverse.

The online version of this article includes the following source data and figure supplement(s) for figure 10:

**Source data 1.** coordinate files of *Vibrio* AlphaFold models.

**Figure supplement 1.** Overlay of ToxRSp from different *Vibrio* species.

Although the eight-stranded ß-barrel fold of ToxS and VtrC shows similarities with the calycin superfamily (including fatty acid binding proteins FABP and lipocalins; ***Supplementary file 1f***; ***Kinch et al., 2022***; ***Tomchick et al., 2016***), the formation of an obligate heterodimer to fold the barrel including the binding pocket is unique for ToxRSp and VtrAC (***Tomchick et al., 2016***).

Despite the obligate heterodimerization and the structural similarities, another feature of this protein family is the arrangement of a two-gene cassette encoding the two proteins (***Kinch et al., 2022***). Using this information, genetic cluster analysis recently revealed additional members of the VtrAC-like superfamily including BqrP/BqrQ from enteric *Burkholderia pseudomallei*, as well as GrvA/FidL from pathogenic *E. coli* O157:H7 Sakai strain (***Kinch et al., 2022***).

The distinctive structural and functional features of ToxRS and VtrAC support a common superfamily of co-component signalling systems of VtrAC-like protein complexes, whose sensory function is strictly linked to dimerization and ligand binding (***Kinch et al., 2022***). So far VtrAC and ToxRSp remain the only experimentally solved structures of this relatively new defined superfamily.

## ToxRSp obligate dimer formation is conserved among *Vibrio* strains

The predicted protein structures of ToxRSp proteins from different *Vibrio* strains (***Supplementary file 1e***) share the distinctive ToxRSp fold consisting of a ß-barrel ToxSp and an αß folded ToxRp (***Figure 10***, Fig. ***Figure 10—figure supplement 1***). Although ToxRp structure varies among the strains, the barrel shaped lipocalin-like structure of ToxSp is consistent, clearly visible in the alignment of all structures (***Figure 11***, ***Figure 11—figure supplement 1***, ***Figure 11—figure supplement 2***).

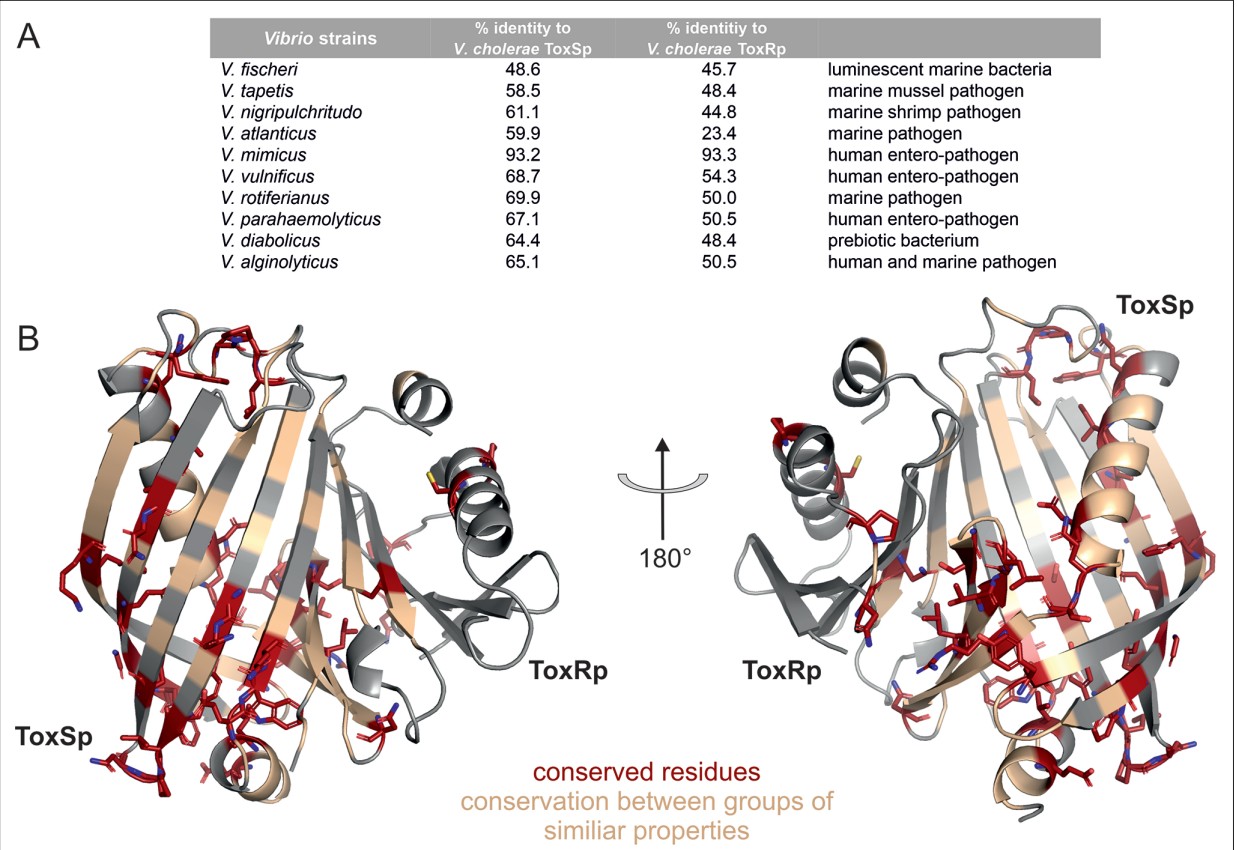

| *Vibrio* strains | % identity to *V. cholerae* ToxSp | % identitiy to *V. cholerae* ToxRp | |
|---|---|---|---|
| *V. fischeri* | 48.6 | 45.7 | luminescent marine bacteria |
| *V. tapetis* | 58.5 | 48.4 | marine mussel pathogen |
| *V. nigripulchritudo* | 61.1 | 44.8 | marine shrimp pathogen |
| *V. atlanticus* | 59.9 | 23.4 | marine pathogen |
| *V. mimicus* | 93.2 | 93.3 | human entero-pathogen |
| *V. vulnificus* | 68.7 | 54.3 | human entero-pathogen |
| *V. rotiferianus* | 69.9 | 50.0 | marine pathogen |
| *V. parahaemolyticus* | 67.1 | 50.5 | human entero-pathogen |
| *V. diabolicus* | 64.4 | 48.4 | prebiotic bacterium |
| *V. alginolyticus* | 65.1 | 50.5 | human and marine pathogen |

**Figure 11.** Sequential and structural analysis of sensory domains of ToxRS from different *Vibrio* strains. ToxRSp from different *Vibrio* species exhibit similar heterodimer formations. Especially ToxSp ß-barrel structure, which forms the binding cavity, seems to be conserved. (**A**) Sequential analysis of conserved residues of ToxRSp. Sequential alignment was done using ToxRS amino acid sequences from *Vibrio* species (***Supplementary file 1e***). The sequence identity (%) of ToxR and ToxS proteins from different strains to *V. cholerae* is shown in the table above the structures. (**B**) Conserved residues are highlighted on the ToxRSp structure. Conserved residues are coloured in red and shown in sticks, residues which are conserved between groups of similar properties are coloured in gold. Most conserved regions could be found in ToxSp, especially at the openings of the barrel. ToxRp seems to have a higher sequential variability.

The online version of this article includes the following figure supplement(s) for figure 11:

**Figure supplement 1.** Sequential analysis of ToxSp proteins from different *Vibrio* species.

**Figure supplement 2.** Sequential analysis of ToxRp proteins from different *Vibrio* species.

Sequence alignments (***Figure 11—figure supplement 1***, ***Figure 11—figure supplement 2***) reveal numerous conserved residues of ToxSp compared to more diverse ToxRp (***Figure 11***). Most of the conserved residues of ToxSp are located at the openings of the barrel as well as ß strands 8, 7 and 6 which are involved in ToxRp interaction (***Figure 11***). Also, residues near the N-terminus, which is probably located near the membrane, seem to be highly conserved in *Vibrio* species. The aspartate residue which is involved in the intermolecular salt bridge with ToxRp is also one of the conserved residues of ToxSp (***Figure 10—figure supplement 1***). The high content of conserved residues in the binding pocket of ToxSp proposes that its general function is to bind to small hydrophobic molecules similar to bile acids.

Compared to ToxSp, ToxRp sequence seems to be only conserved between human colonizing strains (***Figure 11—figure supplement 2***). Most of ToxRp conserved residues are involved in the formation of the hydrophobic core, as well as the ToxSp interface (***Figure 11***). Additionally, two cysteine residues are conserved, as well as an arginine/lysine residue involved in the formation of the intermolecular salt bridge with the conserved aspartate residue of ToxSp. ToxRp residues involved in the binding to S-CH are also conserved ($Val283_{ToxRp}$, $Arg281_{ToxRp}$).

In conclusion, the sequential and structural alignments reveal that ToxSp is conserved among different *Vibrio* species, whereas ToxRp shows significant sequence similarities only among human colonizing strains. The high number of conserved residues involved in the direct interaction with bile acid furthermore proposes that ToxRSp complex has a common sensory function in *Vibrio*'s.

## Conclusion

Cholera represents a dangerous disease which leads to sudden epidemic outbreaks due to the perseverance of the causative *V. cholerae* in aqueous environments (*Ahmed and Nashwan, 2022*; *Ali et al., 2015*). The bacterium has an immense ability to adapt to challenging conditions due to its sensitive sensory systems. Herein, we present insights into the regulatory bile binding protein complex of *V. cholerae* involving sensory proteins ToxR and ToxS. Our analysis reveals ToxS as a conserved environmental sensor in *Vibrio* strains, functionally effective only in complex with transcription factor ToxR. Targeting the disruption of this vital binding mechanism provides a powerful tool for inhibiting *Vibrio*'s adaption to its host and consequently its virulent action.

## Materials and methods

### Protein expression and purification

All constructs listed in table *Supplementary file 1a* were generated by using standard procedures and verified via automated sequencing. For protein expression *E. coli* BL21 DE cells were used. Cells were grown under shaking at 180 rpm and induced with 1 mM IPTG when the $OD_{600}$ reached 0.6–0.8. ToxRp producing cultures were incubated at 37 °C overnight, ToxSp was expressed at 37 °C before induction, after growth at 20 °C overnight. For production of non-isotopically labelled proteins LB media was used. For isotopically labelled proteins cells were grown in M9 minimal media containing $^{15}$N-labelled $(NH_4)_2SO_4$ and $^{13}$C labelled glucose as the sole nitrogen and carbon sources, respectively.

After overnight expression, cell cultures were centrifuged, and the pellet dissolved in 20 ml of loading buffer containing either 8 M urea, 300 mM sodium chloride, 10 mM imidazole pH 8 for ToxRp, or 20 mM Tris, 300 mM sodium chloride, 10 mM imidazole pH 8 for ToxSp. Protease inhibitor mix (Serva, Protease Inhibitor Mix HP) was added to the loading buffer. The cells were disrupted via sonication. The lysate was centrifuged, and the supernatant was loaded on a gravity column containing 2 ml of Ni-NTA agarose beads. The column was washed with 15 column volumes (CV) of loading buffer followed by 5 CV of the loading buffers containing 1 M sodium chloride and 5 CV of the loading buffer containing 20 mM imidazole. His-tagged proteins were eluted with 5 CV elution buffer containing 330 mM imidazole. ToxRp constructs were refolded overnight by dialysis at 4 °C in 50 mM $Na_2HPO_4$, 300 mM sodium chloride, pH 8. The final purification step includes purification by FPLC using a HiLoad 26/600 Superdex 75 pg column in 50 mM sodium phosphate, 300 mM sodium chloride, pH 8.

### Crystallization and data acquisition

Periplasmic protein domains were expressed separately and added in a 1:2 ratio, with excess of ToxSp. Due to ToxSp instability saturation of ToxRp binding sites is only achieved when ToxSp is added in two molar excess. Unbound ToxSp was eliminated by SEC. The protein complex was dialyzed against crystallization buffer (20 mM Tris, 150 mM NaCl, 0.02% $NaN_3$, pH 8) and concentrated to 30 mg/ml using an Amicon Ultra-15 Centricon (Millipore Merck, Darmstadt, Germany). Crystallization experiments were performed with ORYX8 pipetting robot (Douglas Instruments, Hungerford, UK) and sitting drop vapor-diffusion method using 96-well 3-drop plates (SwissCI AG, Neuheim, Switzerland). Initial crystallizations were setup using commercially available screens: Index (Hampton Research, United States), JCSG +and Morpheus (Molecular Dimensions, United states). Each drop was set up with 0.5 μl protein mix and 0.5 μl of the screen condition. Crystallization plates were incubated at 16 ° C and H11 condition JCSG +containing 0.2 M magnesium chloride hexahydrate, 0.1 M Bis-Tris pH 5.5 and 25 % w/v PEG3350 yielded the best diffracting single crystals after two months. The crystals were stored in liquid nitrogen until screening and data collection at the ID30A-3 beamline at the ESRF (Grenoble, France) using a PILATUS detector. Data were collected at 100° K at 0.96770 Å wavelength with 0.20° oscillation range and a total of 700 images.

Data were processed using XDS (*Kabsch, 2010*) and AIMLESS (v.0.7.7.) (*Evans and Murshudov, 2013*). Crystals diffracted up to 3 Å and belong to the space group P6$_5$ with cell constants of 72.5 Å,

72.5 Å, 79.4 Å and 90°, 90° and 120°. The structure was solved with molecular replacement using Phaser (2.8.3) (*McCoy et al., 2007*). A partial ab initio model of the assembled complex of ToxSp and ToxRp predicted by AlphaFold-Multimer (*Evans et al., 2021*) was used as a template. Employing the complete predicted model of the complex for phasing failed, hence only the larger part of it, ToxSp, was used for initial phasing with Phaser. Furthermore, the predicted model of the ToxSp part was trimmed down at parts below 0.7 predicted IDDT score. The trimmed partial model of ToxSp was suitable to position the model correctly, solve the phase problem and obtain phases that were sufficient enough to place the second molecule, ToxRp, using Phaser. The solved structure was rebuilt and improved in Coot (v.0.9.6.) (*Emsley et al., 2010*) and with ChimeraX (v.1.3.) (*Pettersen et al., 2021*) using the ISOLDE plugin (*Croll, 2018*). Waters were placed manually within Coot. Refinement was performed with REFMAC5 (*Murshudov et al., 2011*). The final refined model was analysed and validated with PISA (*Krissinel and Henrick, 2007*), MolProbity (*Chen et al., 2010*) and PDB. Data collection and refinement statistics (*Supplementary file 1b*) were generated within Phenix Table1 (*Liebschner et al., 2019*). The structure was deposited at the PDB with the DOI: https://doi.org/10.2210/pdb8ALO/pdb and the PDB accession code 8ALO.

### *Ab initio* models of ToxRp, ToxSp and dimers with AlphaFold

*Ab initio* models of ToxRp and ToxSp (*V. cholerae*) were calculated using an AlphaFold 2.1 (*Jumper et al., 2021*) installation with full databases in standard configuration for procaryotes. The search models used for molecular replacement of the heterodimer ToxRSp (*V. cholerae*) as well as the models of the proposed homodimer ToxSp (*V. cholerae*) were calculated on an AlphaFold-Multimer 2.1 (*Evans et al., 2021*) installation with full databases in standard configuration for procaryotes.

For calculation of ToxRSp Heterodimers from various *Vibrio* Species shown in *Supplementary file 1e*, an AlphaFold-Multimer 2.2 (*31*) installation in standard configuration with full databases and v2 model weights was used. For all species, 5 models were generated and ranked by the highest model confidence metric (0.8·ipTM + 0.2·pTM).

### NMR experiments

All NMR spectra were recorded on a Bruker Avance III 700 MHz spectrometer equipped with a cryogenically cooled 5 mm TCI probe using z-axis gradients at 25 ° C. All NMR samples were prepared in 90% $H_2O$/10% $D_2O$. The total sample volume was 600 μL. Spectra were processed with NMRPipe (*Delaglio et al., 1995*).

### Diffusion-ordered NMR spectroscopy DOSY

For DOSY experiments all samples were dissolved in the exact same buffer (20 mM Tris, 100 mM NaCl pH 6.5). Sodium cholate hydrate S-CH was added to ToxRSp samples from a 20 mM stock dissolved in the same buffer. The concentration of ToxRSp was 360 μM. The DOSY spectra were recorded using the Bruker dstebpgp3s pulse sequence employing a solvent suppression via presaturation and 3 spoil gradients (*Johnson, 1999*; *Morris and Johnson, 1992*). Relevant parameters include a gradient duration of 1.4ms, a diffusion time of 70ms and a linear 32-step gradient ramp profile from 1.06 to 51.84 G/cm. Due to phase errors in the first gradient steps, the initial measurement points could not be used, thus the series of points was truncated to afford only the stable signal attenuation. The spectra were processed with Bruker Dynamics Center using standard parameters.

### Saturation transfer difference (STD)

Samples for STD experiments (*Mayer and Meyer, 1999*) contained an excess of S-CH compared to protein (protein to ligand ratios: ToxRp 1:50, ToxRSp 1:20, ToxSp 1:20). Protein concentrations of samples were: 50 μM ToxRp, 300 μM ToxRSp, 30 μM ToxSp. Three protein selective regions were chosen for saturation: 6000 Hz (amide region), 5000 Hz (amide region) and –5000 Hz (negative control). Experiments with ToxRp and bile were done in two buffers: 20 mM Tris, 100 mM NaCl pH 6.5. Reference spectrum of S-CH was recorded in the exact same buffer at a concentration of 2 mM. Additionally, experiments were repeated in buffer 20 mM Kpi, 200 mM NaCl, pH 8 according to previously published NMR experiments (*Midgett et al., 2017*).

### NMR titration experiments

Titration experiments with ToxRSp (360 μM) and ToxRp (200 μM) were performed in 50 mM $Na_2HPO_4$, 100 mM NaCl at pH 6.5. Additionally, experiments were repeated in buffer 20 mM Kpi, 200 mM NaCl,

pH 8 according to previously published NMR experiments (*Midgett et al., 2017*). S-CH was titrated using a 20 mM stock solution prepared in the exact same buffer. 1D $^1$H proton spectra as well as 2D $^1$H-$^{15}$N-HSQC spectra (*Davis et al., 1992*) were recorded for each step of the titrations. Due to the disappearance of ToxRSp peaks upon bile addition, bile interaction experiments were performed using low amounts of S-CH. Following protein-to-ligand ratios of S-CH were used for ToxRSp experiments: 0.1, 0.2, 0.3, 0.4, 0.5, 0.7, 0.8, 1, 1.5, 2, 4. For *Figure 7* only ratios 0.3, 0.5, and 0.7 are shown. Regarding ToxRp-bile titration following protein-to-ligand ratios of S-CH were used: 1, 3, 13, 50, 100, 150, 200, shown in *Figure 8—figure supplement 1*.

## CSP analysis

Spectra were processed with NMRPipe (NMRDraw v5.6 Rev) (*Delaglio et al., 1995*) and analyzed with CcpNmr Analysis 2.4.2. (*Skinner et al., 2016*). Molecular images were created with PyMOL (v2.0 Schrödinger, LLC). For ToxRp NMR experiments published NMR assignments were used (*Gubensäk et al., 2021b*).

Euclidean distances, also called *d*-values, were calculated as described by *Williamson, 2013*, (*Becker et al., 2018*):

$$d = \sqrt{\frac{1}{2}[\delta_H^2 + (\alpha * \delta_N)^2]} \tag{1}$$

$d$…Euclidean distance, $\delta_N$…$^{15}$N chemical shift changes, $\delta_H$…$^1$H chemical shift changes, $\alpha$…scaling factor (glycines $\alpha$= 0.2, all other amino acids $\alpha$= 0.14)

A threshold value was determined according to the procedure described by *Schumann et al., 2007* to exclude residues with non significant chemical shift changes.

The dissociation constant ($K_d$) was calculated for each amino acid individually using CcpNmr Analysis 2.4.2. (*Skinner et al., 2016*) and the following equation:

$$\Delta\delta_{obs} = \Delta\delta_{max} \left\{ ([P]_t + [L]_t + K_d) - \sqrt{([P]_t + [L]_t + K_d)^2 - 4[P]_t L_t}/2[P]_t \right. \tag{2}$$

$\Delta\delta_{obs}$…change in observed shift, $\Delta\delta_{max}$…maximum shift change on saturation, $[P]_t$…total protein concentration, $[L]_t$…total ligand concentration, $K_d$…dissociation constant

## Size exclusion chromatography (SEC)

SEC experiments were carried out on an ÄKTA pure 25 (Cytavi, Marlborough, USA) using a Superdex 200 Increase 10/300 column (Cytiva) with a flowrate of 0.4 ml/min. As sample and running buffer a phosphate buffer (50 mM $Na_2HPO_4$, 300 mM NaCl, pH 8, supplemented with 0.02% $NaN_3$) was used. A total of 100 µl of ToxRS without and with bile acid S-CH were measured. Sample 1 was 1 mM of ToxRSp complex, sample 2 was 300 µM of ToxRSp with equimolar amount (1:1) of S-CH.

## Microfluidic modulation spectroscopy (MMS)

For initial MMS measurements three 1 ml ToxRSp samples of 5 mg/ml were prepared in 50 mM $Na_2HPO_4$, 300 mM NaCl pH 8.0. A 20 mM stock of S-CH was made in the exact same buffer and added to the protein complex for final protein to ligand ratios of 1:1 and 1:2. Also, a ToxRSp sample without ligand was measured. Results are shown in *Figure 4*. For each sample a reference buffer, containing the same amount of S-CH was used.

Additionally, measurements with increasing protein-to-ligand ratios were performed with 4.5 mg/ml ToxRSp complex in 50 mM $Na_2HPO_4$, 300 mM NaCl pH 8. A 20 mM stock of S-CH was made in the exact same buffer and added to the protein complex for final protein to ligand ratios of: 1:0, 1:1, 1:2, 1:3, 1:5, 1:20. For each sample, a reference buffer, containing the same amount of S-CH was used. Results are shown in *Figure 4—figure supplement 1*.

Samples were measured and analysed with a RedShiftBio AQS3pro MMS production system equipped with sweep scan capability (RedShiftBio, Boxborough, MA, USA). For background subtraction, chemically identical buffer and buffer-bile mixtures were loaded pairwise with the corresponding sample onto a 96-well plate. The instrument was run at a modulation frequency of 1 Hz and with a microfluidic transmission cell of 23.5 µm optical pathlength. The differential absorbance spectra of the sample against its buffer reference were measured across the amide I band (1714–1590 cm-1). For

each spectrum, triplicate measurements were collected and averaged. The data were analysed on the RedShiftBio *delta* analysis software.

## SEC-SAXS

SEC-SAXS was performed at the EMBL P12 bioSAXS beamline (PETRA III, Hamburg, Germany). Samples were dissolved in 50 mM $Na_2HPO_4$, 300 mM NaCl pH 8.0 and 3% glycerol to prevent radiation damage. For the single proteins, ToxSp (10 mg/ml), ToxRp (10 mg/ml) and ToxRSp (14 mg/ml), a Superdex 75 Increase 10/300 column was used. For the ToxRS complex in the presence of bile acid S-CH (10 mg/ml), a Superdex 75 Increase 5/150 column was used. The column outlet was connected to the flow-through SAXS cell.

Throughout the elution, frames (I(q) vs q, where q=4πsinθ/$\lambda$ , and 2θ is the scattering angle) were collected successively. The data were normalized to the intensity of the transmitted beam and radially averaged; the scattering of the solvent was subtracted from the sample frames using frames corresponding to the scattering of the solvent. For optimized selection of buffer and sample frames the program CHROMIXS was employed. Data were further analysed and prepared via PRIMUS from the ATSAS software (*Manalastas-Cantos et al., 2021*), model preparation was done using DAMMIF at the online ATSAS server provided by EMBL Hamburg (https://www.embl-hamburg.de/biosaxs/atsas-online/). Comparison to theoretical scattering curves was performed with CRYSOL. Figures were done using PyMOL (*Schrodinger, 2010*).

The data have been deposited in the SASBDB databank. Accession codes: SASDR25, SASDR35, SASDR45, SASDR55.

## MD simulations

The Cartesian coordinates of the complex between ToxRp and ToxSp were used as initial structure for our MD simulations. The minimum energy structure of the ligand cholate was first of all minimized with the GFN2-xTB (*Bannwarth et al., 2019*) using CREST, (*Grimme, 2019*), minimized with r²SCAN (*Furness et al., 2020*) and the RESP point charges for all atoms of cholate were derived from the electrostatic potential computed at the HF-6–31 G* level of theory with Gaussian16 v. C. (*Gaussian 16 and Revision B.01, 2016*). The protonation states of the titritable residues on the protein complex were calculated via the H++ web server (*Anandakrishnan et al., 2012*; *Gordon et al., 2005*; *Myers et al., 2006*) assuming a pH value of 8.0. Cholate was manually docked at the hydrophobic inner chamber of the ToxRSp complex guided by the superimposition of the crystallographic structure of *Vibrio parahaemolyticus* VtrA/VtrC complex in the presence of taurodeoxycholate (PDB id. 5KEW) (*Alnouti, 2009*; *Li et al., 2016*).

MD simulations were carried out using the suite of programs AMBER20 (*Amber20, 2022*) Protein residues and solvated ions were treated with the AMBER ff19SB force-field (*Ponder and Case, 2003*) and cholate was described as AMBER atoms types following the standard procedure in antechamber. We simulated two systems: the apo complex (ToxRSp) and the cholate-bound complex (ToxRSp +cholate). The two systems were then energetically minimized to avoid close contacts, and then placed in the center of a cubic box filled with OPC water molecules (*Izadi et al., 2014*). Then, the solvated systems were minimized in three consecutive steps (all protons, solvent and all system) and heated up in 50 ps to from 100 k to 300 K in a NVT ensemble using the Langevin thermostat (gamma friction coefficient of 1.0). Care was taken to constraint the solute during the heating step by imposition of a harmonic force on each atom of the solute of 40 kcal mol⁻¹ Å⁻². Afterwards, these harmonic constraints were gradually reduced up to a value of 10 kcal mol⁻¹ Å⁻² in 4 simulation stages (NVT, 300 K). Then, the systems were switched to constant pressure (NPT scheme, 300 K) and the imposed constraints during the heating step were totally removed. Finally, each of the systems was submitted to three independent MD simulations of 0.8 μs (total simulation time per system: 2.4 μs). Atom-pair distance cut-offs were applied at 10.0 Å to compute the van der Waals interactions and long-range electrostatics by means of Particle-Mesh Ewald (PME) method. SHAKE algorithm was applied to restrain the hydrogen atoms on water molecules. MD trajectory analysis was carried out using the CPPTRAJ (*Roe and Cheatham, 2013*) module from AMBER20 for monitoring the root-mean-square distance (RMSD) and root-mean-square fluctuation (RMSF), amongst other parameters.

## Acknowledgements

We are grateful for the beam time on ID30A-3 at ESRF (Grenoble, France) for intensive diffraction screening and data collection, as well as for staff support during data collection measurements (BAG mx-1740). JR, KZ and TPK acknowledge the support of the field of excellence BioHealth at the University of Graz. We also thank the interuniversity programs NAWI Graz and BioTechMed for financial support. Furthermore, we acknowledge the financial support by Austrian Science Fund FWF for projects: T-1239 for NG, DK W09 for KZ, P 29405 for JR, and doc.fund projects Molecular Metabolism (DOC 50 for TS and KZ) and Biomolecular Structure and Interactions (DOC 130 for MR, KZ and TPK). Financial contributions by the Land Steiermark infrastructure grant "Frontier NMR", project number 1109 are also gratefully acknowledged. Ministerio de Ciencia e Innovación and European Union Regional Development Fund (MICINN/AEI/FEDER/UE) are acknowledged for grant No. PID2021-128751NB-I00 (for IU) and BDG thanks Fundación Martínez Escudero for a Postdoctoral grant.

## Additional information

### Competing interests

Jan Schäfer: is affiliated with Redshift BioAnalytics, Inc which distributes the AQS3pro. Access to the AQS3pro instrument was provided to Nina Gubensäk as part of the RedShiftBio demo lab. The other authors declare that no competing interests exist.

### Funding

| Funder | Grant reference number | Author |
|---|---|---|
| Austrian Science Fund | FWF T-1239 | Nina Gubensäk |
| Austrian Science Fund | FWF DK W09 | Klaus Zangger |
| Austrian Science Fund | FWF P 29405 | Joachim Reidl |
| Land Steiermark | 1109 | Klaus Zangger |
| Ministerio de Ciencia e Innovación and European Union Regional Development Fund (MICINN/AEI/FEDER/UE) | PID2021-128751NB-I00 | Isabel Usón |
| Austrian Science Fund | Biomolecular Structures and Interactions DOC 130 | Tea Pavkov-Keller Markus Rotzinger Klaus Zangger |
| Austrian Science Fund | Molecular Metabolism DOC 50 | Theo Sagmeister Klaus Zangger |
| Fundación Martínez Escudero | Postdoctoral grant | Bruno Di Geronimo |
| University of Graz | | Joachim Reidl Klaus Zangger Tea Pavkov-Keller |

The funders had no role in study design, data collection and interpretation, or the decision to submit the work for publication.

### Author contributions

Nina Gubensäk, Conceptualization, Data curation, Supervision, Funding acquisition, Investigation, Writing - original draft, Project administration, Writing – review and editing; Theo Sagmeister, Data curation, Investigation, Writing – review and editing; Christoph Buhlheller, Lukas Petrowitsch, Investigation; Bruno Di Geronimo, Isabel Usón, Investigation, Writing – review and editing; Gabriel E Wagner, Conceptualization, Data curation, Investigation, Writing – review and editing; Melissa A Gräwert, Jan Schäfer, Pedro A Sánchez-Murcia, Data curation, Investigation, Visualization, Writing – review and editing; Markus Rotzinger, Tamara M Ismael Berger, Investigation, Visualization; Joachim Reidl, Conceptualization, Investigation, Writing – review and editing; Klaus Zangger, Conceptualization, Data curation, Supervision, Funding acquisition, Validation, Investigation, Project administration,

Writing – review and editing; Tea Pavkov-Keller, Conceptualization, Data curation, Supervision, Funding acquisition, Validation, Investigation, Writing - original draft, Project administration, Writing – review and editing

### Author ORCIDs
Nina Gubensäk http://orcid.org/0000-0002-0415-4299
Gabriel E Wagner https://orcid.org/0000-0002-5704-3955
Markus Rotzinger http://orcid.org/0000-0002-0411-3403
Tea Pavkov-Keller http://orcid.org/0000-0001-7871-6680

### Decision letter and Author response
Decision letter https://doi.org/10.7554/eLife.88721.sa1
Author response https://doi.org/10.7554/eLife.88721.sa2

---

## Additional files

### Supplementary files
• Supplementary file 1. Supplementary tables for 'Vibrio cholerae's ToxRS Bile Sensing System'. (**a**) Description of expression constructs. List of plasmids and amino acid sequences used for the expression of ToxRp and ToxSp. (**b**) Crystal data and structure refinement table of ToxRSp. (**c**) Analysis of size-exclusion chromatography coupled with solution small-angle X-ray scattering SEC-SAXS. According to SAXS curves, ToxRp is monomeric in solution, whereas ToxSp seems to form dimers. Upon bile addition, the radius of gyration RG, the maximum particle size Dmax and the molecular weight of the ToxRSp complex increases. (**d**) NMR derived dissociation constants for ToxRp and sodium cholate hydrate. NMR titration experiments reveal a weak binding of bile to ToxRp with a dissociation constant of 2.6 mM. Table S5 includes ToxRp residue number, chemical shift distance, dissociation constant Kd [M], error of Kd. The calculated average Kd is 2.7±1.4 mM. (**e**) List of amino acid sequences of sensory domains of ToxRS proteins from different Vibrio species. Table S5 includes UniProt ID of the used sequences as well as sequence identity values to ToxRSp from *V. cholerae*. (**f**) Structural homology search of ToxSp bound to ToxRp using Dali server (*Holm, 2022*).

• Supplementary file 2. ToxRSp atom-atom interactions. List of all atom interactions of ToxRSp (pdb: 8ALO). The list was created using PDBsum online server (*Laskowski et al., 2018*).

• MDAR checklist

### Data availability
Diffraction data have been deposited in PDB under the accession code 8ALO. SAXS data have been deposited: ToxR - SASDR25, ToxS - SASDR35, ToxR:ToxS - SASDR45, ToxR:ToxS:bile - SASDR55. All data generated or analysed during this study are included in the manuscript and supporting files.

The following datasets were generated:

| Author(s) | Year | Dataset title | Dataset URL | Database and Identifier |
|---|---|---|---|---|
| Gubensaek N, Sagmeister T, Pavkov-Keller T, Zangger K, Bulheller C, Wagner GE | 2023 | Heterodimer formation of sensory domains of Vibrio cholerae regulators ToxR and ToxS | https://www.rcsb.org/structure/8ALO | RCSB Protein Data Bank, 8ALO |
| Graewert M | 2023 | Periplasmic domain of cholera toxin transcriptional activator ToxR | https://www.sasbdb.org/data/SASDR25/ | SASBDB, SASDR25 |
| Graewert M | 2023 | Periplasmic domain of cholera transmembrane regulatory protein ToxS | https://www.sasbdb.org/data/SASDR35/ | SASBDB, SASDR35 |

*Continued on next page*

*Continued*

| Author(s) | Year | Dataset title | Dataset URL | Database and Identifier |
|---|---|---|---|---|
| Graewert M | 2023 | A complex between the periplasmic domains of cholera toxin transcriptional activator ToxR and transmembrane regulatory protein ToxS | https://www.sasbdb.org/data/SASDR45/ | SASBDB, SASDR45 |
| Graewert M | 2023 | A complex between the periplasmic domains of cholera toxin transcriptional activator ToxR and transmembrane regulatory protein ToxS bound to bile salt | https://www.sasbdb.org/data/SASDR55/ | SASBDB, SASDR55 |

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
