## [Editor Report]

This study provides important insights into the structure and mechanism of the sensory protein complex ToxR/S that is associated with the survival and virulence of the cholera pathogen. The structural studies are solid and supported by a series of biophysical experiments revealing a split, periplasmic protein binding interface for bile acid. Results are of interest to protein biochemistry and pharmacology where they may open new routes for the treatment of cholera disease.

---

## [Decision Letter]

**Decision letter after peer review:**

Thank you for submitting your article "*Vibrio cholerae*'s ToxRS Bile Sensing System" for consideration by *eLife*. Your article has been reviewed by three peer reviewers, including Hannes Neuweiler as Reviewing Editor and Reviewer #1 and the evaluation has been overseen by Amy Andreotti as the Senior Editor.

The reviewers have discussed their reviews with one another, and the Reviewing Editor has drafted this letter to help you prepare a revised submission.

The reviewers support the publication of your work as an article in *eLife* after their concerns have been addressed. Please address all concerns in a point-by-point response letter and a revised manuscript. The reviews and the concerns (detailed in the sections "recommendations for the authors") are below.

To improve the quality of your manuscript further, the reviewers suggest that you use the space provided by *eLife* (no limits on the number of display items) and move Figures that show important results from biophysical experiments (i.e., SEC-MALS, NMR, and ITC) from Supporting Information into the main article, together with improved Figure legends.

*Reviewer #1 (Recommendations for the authors):*

1) SEC-MALS experiments show an unreasonable increase of molecular weight of the ToxRSp complex by ~10 kDa upon binding of bile acid in 1:1 stoichiometry (Figure S11). On a side note: there is a mistake in the labelling of panels in Figure S11. Panel A is labelled ToxRSp but MALS data show a Mw of ~40 kDa. Panel B is labelled ToxRSp:bile acid and MALS data show a Mw of ~30 kDa, which is not reasonable. On page 6 the authors argue for additional binding patches that would explain the 10-kDa increase of Mw. But this is unreasonable from a 1:1 stoichiometry in the SEC-MALS experiment. Bile acid has a Mw of ~0.5 kDa. The interactions seen in MD simulations likely indicate transient, rather non-specific interactions and not specific binding sites. The 10-kDa increase of Mw in SEC-MALS experiments remains ambiguous. The shape of the peak in the chromatogram in panel B is not symmetric. A possible explanation could be sample heterogeneity induced by formation of the ternary complex, which would require further investigations and explanations.

2) With regard to point 1): Can the authors comment on the concentration of bile acid in the stomach, which has implications on their findings? The concentration is of relevance to the discussion regarding weak bile-ToxRSp interactions showing high mM Kd values.

3) The readability of the manuscript would benefit from putting some of the supplementary figures into the main manuscript. E.g., it would be helpful to show Figures S2, S3 and S5 next to main Figure 1. They address the interesting disorder-order transition and details on molecular interactions within the interface, which are described to on page 3-4.

4) Can the authors infer the order of events in the assembly of the ToxRSp-bile complex from their data? From the structure it looks like that the folding-on-binding interaction of ToxRp cages the ligand, which would indicate that ToxRp binds after or in cooperation with formation of a ToxSp-bile complex. The structure seems to show that the flexible b1/b2 loop folds over the binding pocket. Could this explain the observed 14% increase of disorder upon bile binding in MMS experiments (Figure 3)?

5) Can the authors expand their discussion on why a split binding interface for bile acid evolved? What is the benefit or the rational? Couldn't a conventional single-domain periplasmic binding interface do the same job? The recruitment of ToxS to ToxR seems to be a requirement for successful signal transduction.

*Reviewer #3 (Recommendations for the authors):*

Although the structural data are very interesting, some phenotypic assays with protein mutants will significantly enhance the significance of the study.

The provided biophysical experiments to explain the binding mechanism, raise more questions than answers. The data presented in the Supplementary Information section, are not convincingly supported. Below are some points regarding the interaction study.

– SEC-MALS: The difference of elution volume between the bound- and free ToxRSp (16.4 vs 16.3 ml) is too small to be significant. The elution peak in the presence of bile acid is not symmetrical, suggesting a mixture between a bound and a free-state. Did the author try to add more bile acid in order to saturate the complex and have a more significant shift of the elution volume?

– Figure S10: The ITC data appear to be inconsistent. How were the data fitted? The heat of interaction shows a negative value, whereas the fitted enthalpy is positive. The stoichiometry of 0.5 should be explained. Additionally, could the authors show the raw ITC data?

– Figure S14: The amount of added sodium cholate seems to be very high. How can we be certain that the spectral modifications are not caused by changes in viscosity? Have the authors attempted to add another hydrophobic compound within a similar concentration range?

How are the chemical shift perturbations (CSP) considered? Typically, CSP values greater than 1 or 2 times the standard deviation of the shift of all residues are significant.

– It would be helpful to include the 1D NMR spectra of Sodium cholate. This would help in understanding the STD experiments and Figure S8A. In Figure S8A, the shift observed in the 1D experiment may be due to a variation in pH. If the protein signals disappear in the 2D HSQC due to the very large molecular weight, as explained by the authors, why is the amide region still well-visible in the 1D experiment?

Figure 1: Regarding the sentence, "ToxSp ß8 position is stabilized by minor main chain hydrogen bonds," what is meant by "minor HBonds"?

Figure 2: Based on the figure, the conformational changes in the loops do not appear to be significant, as mentioned in the text (just above, line 16). Could you please highlight the conformational changes more clearly or provide a zoomed-in view?

Page 5, line 3: The sentence, "The conformation of ß4 shows only slight changes when interacting with ß5 or its aqueous environment," is not clear. Could you please specify what the slight conformational changes are?

Figure S6A and other figures, the legend needs to be completed, please explain different panels.

Figure 4: Panel C is not clear.

Overall, the figure legends and quality of the figures are currently not suitable for publication.

References: The references for Gaussian and Amber are not well formatted.

---

## [Author Response]

Reviewer #1 (Recommendations for the authors):1) SEC-MALS experiments show an unreasonable increase of molecular weight of the ToxRSp complex by ~10 kDa upon binding of bile acid in 1:1 stoichiometry (Figure S11). On a side note: there is a mistake in the labelling of panels in Figure S11. Panel A is labelled ToxRSp but MALS data show a Mw of ~40 kDa. Panel B is labelled ToxRSp:bile acid and MALS data show a Mw of ~30 kDa, which is not reasonable. On page 6 the authors argue for additional binding patches that would explain the 10-kDa increase of Mw. But this is unreasonable from a 1:1 stoichiometry in the SEC-MALS experiment. Bile acid has a Mw of ~0.5 kDa. The interactions seen in MD simulations likely indicate transient, rather non-specific interactions and not specific binding sites. The 10-kDa increase of Mw in SEC-MALS experiments remains ambiguous. The shape of the peak in the chromatogram in panel B is not symmetric. A possible explanation could be sample heterogeneity induced by formation of the ternary complex, which would require further investigations and explanations.

We appreciate the remark about the switched molecular weight assignment and agree with reviewer 1 that the molecular weight increase (10 kDa) determined by SEC-MALS experiments of ToxRSp with bile acid remains ambiguous. Our understanding is that bile acid may interfere with the absorption properties of the complex. Also, the interaction of bile acid with aromatic residues in the binding pocket of ToxRSp could induce a quenching effect resulting altogether in an incorrect normalization of the light scattering signal. Therefore, we decided to replace the graph and show the UV traces instead of the light scattering, which reveal clear differences between the samples with and without bile acid indicating an interaction between ToxRSp and bile acid. We agree with reviewer 1 about the non symmetrical shape of the ToxRSp-bile peak and agree about the possibility of a transient ternary complex formation. To gain more clarity, we performed additional experiments (NMR and MMS titrations) further supporting the entangled dynamic mechanism of ToxRSp bile sensing but excluding the hypothesis of the formation of (stable) bile induced ToxRSp multimers.

Although we performed multiple interaction experiments using several biophysical methods we cannot provide a clear explanation for the increase of size of ToxRSp upon bile addition. Instead we could show that ToxRSp and bile acid are interacting according to a complex scheme involving a binding cavity and surface attachment of bile acid molecules. The exact impact of bile acid on ToxRSp ternary structure remains to be elucidated.

Still, the experiments reveal an entangled concentration dependent bile sensing mechanism of ToxRSp involving subtle changes of ToxRSp secondary structure which may be relevant for transcriptional activity.

2) With regard to point 1): Can the authors comment on the concentration of bile acid in the stomach, which has implications on their findings? The concentration is of relevance to the discussion regarding weak bile-ToxRSp interactions showing high mM Kd values.

We appreciate the remark about the bile concentrations in the human gut and added a section on this topic to the results and discussion chapters.

3) The readability of the manuscript would benefit from putting some of the supplementary figures into the main manuscript. E.g., it would be helpful to show Figures S2, S3 and S5 next to main Figure 1. They address the interesting disorder-order transition and details on molecular interactions within the interface, which are described to on page 3-4.

We agree with reviewer 1 and added new figures (now: Figure 2 and Figure 3) to the manuscript.

4) Can the authors infer the order of events in the assembly of the ToxRSp-bile complex from their data? From the structure it looks like that the folding-on-binding interaction of ToxRp cages the ligand, which would indicate that ToxRp binds after or in cooperation with formation of a ToxSp-bile complex. The structure seems to show that the flexible b1/b2 loop folds over the binding pocket. Could this explain the observed 14% increase of disorder upon bile binding in MMS experiments (Figure 3)?

Experiments were performed with already established ToxRSp complexes. Bile acid was added after the ToxRSp complex formation. Nevertheless, we cannot rule out that the mentioned folding-on-binding interaction happens *in-vivo*. We appreciate the notion of reviewer 1 and included this topic in the Discussion section.

We agree with reviewer 1 about the importance of conformational changes upon bile interaction and added a new figure (Figure 5C) with a more detailed discussion regarding this topic. The conformational changes of ß1/ß2 may be an explanation for the increase of disorder, but since also other regions seem to be affected e.g. helical regions, bile acid binding could also influence the stability of helices. This remains to be elucidated.

5) Can the authors expand their discussion on why a split binding interface for bile acid evolved? What is the benefit or the rational? Couldn't a conventional single-domain periplasmic binding interface do the same job? The recruitment of ToxS to ToxR seems to be a requirement for successful signal transduction.

We agree with reviewer 1 that the formation of the complex is necessary for bile sensing and a single protein would be more efficient. We expanded the discussion accordingly.

Reviewer #3 (Recommendations for the authors):Although the structural data are very interesting, some phenotypic assays with protein mutants will significantly enhance the significance of the study.

We agree with reviewer 3 that phenotypic assays with ToxRSp mutants would be an interesting experiment, which we are considering for future studies. Nevertheless, mentioned experiments are time consuming and deliver considerable amount of data which would implicate another separate publication. We believe that the experiments shown in this manuscript already provide crucial and detailed information about ToxRS which we would like to make accessible for the scientific community.

The provided biophysical experiments to explain the binding mechanism, raise more questions than answers. The data presented in the Supplementary Information section, are not convincingly supported. Below are some points regarding the interaction study.– SEC-MALS: The difference of elution volume between the bound- and free ToxRSp (16.4 vs 16.3 ml) is too small to be significant. The elution peak in the presence of bile acid is not symmetrical, suggesting a mixture between a bound and a free-state. Did the author try to add more bile acid in order to saturate the complex and have a more significant shift of the elution volume?

We agree with reviewer 3 that the difference in the elution volume between ToxRSp with and without bile acid is rather small, but in regard to the molecular weight increase observed by other methods (NMR, SAXS) we believe it is still relevant to mention. Binding of bile acid to ToxRSp induces a broadening of the ToxRSp SEC peak and a change of its shape indicating a binding event.

We agree with reviewer 3 about the non symmetrical shape of the ToxRSp-bile peak and agree with the possibility of a mixture of bile-induced states of ToxRSp. But since higher bile to protein ratios did not influence the values significantly, we assume that peak broadening occurs due to the complex dynamic mechanism of ToxRSp bile sensing involving a binding cavity and surface attachment of bile acid molecules. Although we performed multiple interaction experiments using several biophysical methods we cannot offer a clear explanation about the impact of bile acid on the ternary structure of ToxRSp. Instead we could show that ToxRSp bile sensing involves concentration dependent distinctive conformational changes which could be crucial for ToxR transcriptional activity.

– Figure S10: The ITC data appear to be inconsistent. How were the data fitted? The heat of interaction shows a negative value, whereas the fitted enthalpy is positive. The stoichiometry of 0.5 should be explained. Additionally, could the authors show the raw ITC data?

We agree that ITC data are inconsistent therefore we performed additional experiments which we present in the revised manuscript. ITC was the first experiment we performed for investigating ToxRSp bile sensing. Nevertheless, it turned out bile binding to ToxRSp is more complex than expected. To obtain preliminary information from the ITC we performed analysis with the single site independent model. However, since ITC data seems to raise more questions than it answers, we decided to remove the data from the manuscript and concentrate on the results obtained by other more detailed biophysical characterizations. Additionally, we added further experiments (NMR, MMS) supporting our theory of an entangled bile sensing mechanism of ToxRSp.

– Figure S14: The amount of added sodium cholate seems to be very high. How can we be certain that the spectral modifications are not caused by changes in viscosity? Have the authors attempted to add another hydrophobic compound within a similar concentration range?

The concentrations of ToxRSp are relatively high (~350µM) due to the high molecular weight of the complex (30 kDa). Lower concentrations of ToxRSp resulted in a significant decrease of spectra quality. Nevertheless, via NMR experiments (e.g. STD, 1D, CSP, DOSY) in combination with other biophysical methods like SAXS and MMS we could confirm an interaction of ToxRSp with bile acid. Also, we performed additional NMR experiments starting with 0.1 molar excess of bile acid over ToxRSp (corresponding to a final concentration of 35 µM bile acid) resulting in similar outcomes: line broadening and peak shifting. Furthermore, we tested the interaction of ToxRSp with the hydrophobic compound cFP which did not result in line broadening or peak shifting indicating no interaction between ToxRSp and cFP. We included these measurements in the supplementary material (Figure 6 —figure supplement 8).

How are the chemical shift perturbations (CSP) considered? Typically, CSP values greater than 1 or 2 times the standard deviation of the shift of all residues are significant.

The spectrum quality of ToxRSp decreases significantly with increasing amounts of bile acid and therefore hinder a reliable analysis of the chemical shift changes. Also, the high molecular weight of ToxRSp of 30kDa is problematic for NMR experiments, but could be improved by using methods like deuteration and TROSY experiments etc. Nevertheless, spectrum quality still remains a restriction in the analysis of peak shifting. To still gain information about the bile interaction we provide new experiments by which we are able to distinguish between ToxRp and ToxSp signals and could show that ToxSp signals are mainly influenced upon bile interaction indicating a direct interaction.

– It would be helpful to include the 1D NMR spectra of Sodium cholate. This would help in understanding the STD experiments and Figure S8A. In Figure S8A, the shift observed in the 1D experiment may be due to a variation in pH. If the protein signals disappear in the 2D HSQC due to the very large molecular weight, as explained by the authors, why is the amide region still well-visible in the 1D experiment?

We included a spectrum of sodium cholate hydrate in the supplementary information. In case of a change of the pH shift we would expect all signals to be affected. We also made sure to use exactly the same buffer during the NMR titration. The protein was dialyzed several days with multiple changes of the buffer to prevent any pH changes during titration. As shown in the experiments, only some signals experience changes of the chemical shift and the decrease of intensity is also differing between signals as for ToxSp signals the decrease is larger than for ToxRp signals. The line broadening due to increased molecular weight is also visible in 1D experiments. Signal loss due to fast T_2_ relaxation in high molecular weight complexes is much more severe during the much longer pulse-sequences of multi-dimensional experiments, rather than 1D ^1^H spectra.

Figure 1: Regarding the sentence, "ToxSp ß8 position is stabilized by minor main chain hydrogen bonds," what is meant by "minor HBonds"?

We appreciate the notification and rewrote the sentence accordingly.

Figure 2: Based on the figure, the conformational changes in the loops do not appear to be significant, as mentioned in the text (just above, line 16). Could you please highlight the conformational changes more clearly or provide a zoomed-in view?

We provided an additional figure (Figure 5C) comparing the apo and bile-bound state of ToxRSp and discussed it accordingly in the Results section.

Page 5, line 3: The sentence, "The conformation of ß4 shows only slight changes when interacting with ß5 or its aqueous environment," is not clear. Could you please specify what the slight conformational changes are?

We thank reviewer 3 for pointing out this issue. The sentence was rewritten accordingly. There are no significant conformational changes of ToxRp ß4 when interacting with newly formed ß5 (when bound to ToxSp) compared to its conformation when it is exposed to the solution in unbound ToxRp.

Figure S6A and other figures, the legend needs to be completed, please explain different panels.Figure 4: Panel C is not clear.Overall, the figure legends and quality of the figures are currently not suitable for publication.

We corrected the figure legends and generated new figures with higher quality.

References: The references for Gaussian and Amber are not well formatted.

We thank reviewer 3 for the notification and formatted the reference accordingly.